# Structure of dual BON-domain protein DolP identifies phospholipid binding as a new mechanism for protein localisation

Jack Alfred Bryant[1†], Faye C Morris[1†], Timothy J Knowles[1,2†], Riyaz Maderbocus[1,3], Eva Heinz[4], Gabriela Boelter[1], Dema Alodaini[1], Adam Colyer[1], Peter J Wotherspoon[1], Kara A Staunton[1], Mark Jeeves[3], Douglas F Browning[1], Yanina R Sevastsyanovich[1], Timothy J Wells[1], Amanda E Rossiter[1], Vassiliy N Bavro[1], Pooja Sridhar[2], Douglas G Ward[2], Zhi-Soon Chong[5], Emily CA Goodall[1,6], Christopher Icke[1,6], Alvin CK Teo[7], Shu-Sin Chng[5,7], David I Roper[7], Trevor Lithgow[4], Adam F Cunningham[1,8], Manuel Banzhaf[1], Michael Overduin[2,9‡*], Ian R Henderson[1,5‡*]

[1]Institute of Microbiology and Infection, University of Birmingham, Edgbaston, United Kingdom; [2]School of Biosciences, University of Birmingham, Edgbaston, United Kingdom; [3]Institute for Cancer and Genomic Sciences, University of Birmingham, Edgbaston, United Kingdom; [4]Infection & Immunity Program, Biomedicine Discovery Institute and Department of Microbiology, Monash University, Clayton, Australia; [5]Department of Chemistry, National University of Singapore, Singapore, Singapore; [6]Institute for Molecular Bioscience, University of Queensland, St. Lucia, Australia; [7]School of Life Sciences, The University of Warwick, Coventry, United Kingdom; [8]Institute of Inflammation and Immunotherapy, University of Birmingham, Edgbaston, United Kingdom; [9]Department of Biochemistry, University of Alberta, Edmonton, Canada

**\*For correspondence:**
overduin@ualberta.ca (MO);
i.henderson@imb.uq.edu.au (IRH)

[†]These authors contributed
equally to this work
[‡]These authors also contributed
equally to this work

**Reviewing editor:** Sonja V
Albers, University of Freiburg,
Germany

**Abstract** The Gram-negative outer-membrane envelops the bacterium and functions as a permeability barrier against antibiotics, detergents, and environmental stresses. Some virulence factors serve to maintain the integrity of the outer membrane, including DolP (formerly YraP) a protein of unresolved structure and function. Here, we reveal DolP is a lipoprotein functionally conserved amongst Gram-negative bacteria and that loss of DolP increases membrane fluidity. We present the NMR solution structure for *Escherichia coli* DolP, which is composed of two BON domains that form an interconnected opposing pair. The C-terminal BON domain binds anionic phospholipids through an extensive membrane:protein interface. This interaction is essential for DolP function and is required for sub-cellular localisation of the protein to the cell division site, providing evidence of subcellular localisation of these phospholipids within the outer membrane. The structure of DolP provides a new target for developing therapies that disrupt the integrity of the bacterial cell envelope.

## Introduction

Gram-negative bacteria are intrinsically resistant to many antibiotics and environmental insults, which is largely due to the presence of their hydrophobic outer membrane (OM). This asymmetric bilayer shields the periplasmic space, a thin layer of peptidoglycan and the inner membrane (IM). In the model bacterium *Escherichia coli,* the IM is a symmetrical phospholipid bilayer, whereas the OM has a more complex organisation with lipopolysaccharide (LPS) and phospholipids forming an

asymmetric bilayer containing integral β-barrel proteins (*May and Grabowicz, 2018*; *Konovalova et al., 2017*). The OM is also decorated with lipoproteins (approximately 75 have been identified in *E. coli*), many of which, are functional orphans (*Leyton et al., 2012*; *Babu et al., 2006*). Biogenesis of the OM is completed by several proteinaceous systems, which must bypass the peri-plasmic, mesh-like peptidoglycan (*Konovalova et al., 2017*; *Egan, 2018*; *Ekiert et al., 2017*; *Stubenrauch and Lithgow, 2019*). The growth of all three envelope layers must be tightly coordi-nated in order to maintain membrane integrity. Improper coordination can lead to bacterial growth defects, sensitivity to antibiotics, and can cause cell lysis (*Egan, 2018*; *Gray et al., 2015*).

DolP (**d**ivision and **O**M stress-associated **l**ipid-binding **p**rotein; formerly YraP) is a nonessential protein found in *E. coli* and other Gram-negative bacteria (*Goodall et al., 2018*). Loss of DolP results in the disruption of OM integrity, induces increased susceptibility to detergents and antibiotics, and attenuates the virulence of *Salmonella enterica* (*Morris et al., 2018*). Importantly, DolP is a crucial component of the serogroup B meningococcal vaccine where it enhances the immunogenicity of other components by an unknown mechanism (*Bos et al., 2014*). Recently, the *dolP* gene was con-nected genetically to the activation of peptidoglycan amidases during *E. coli* cell division, however this activity has not been directly confirmed experimentally (*Tsang et al., 2017*). In contrast, protein interactome studies suggest DolP is a component of the β-barrel assembly machine (Bam) complex (*Carlson et al., 2019*; *Babu et al., 2018*). While these data suggest that DolP may be involved in outer-membrane protein (OMP) biogenesis and the regulation of peptidoglycan remodeling, its pre-cise function in either of these processes remained unclear. Nonetheless, given its roles in these vital cell envelope processes, and its requirement for virulence and the maintenance of cell envelope integrity, DolP is a potential target for the development of therapeutics.

In this study, we demonstrate that DolP is an outer-membrane lipoprotein functionally conserved amongst Gram-negative bacteria, but with a function distinct from other BON (Bacterial OsmY and nodulation) domain-containing proteins. We solve the NMR solution structure of DolP revealing the first view of a dual BON-domain fold. Extensive structural and functional analyses define a mem-brane:protein interface that binds DolP to anionic phospholipids and provides the basis for a new mechanism for targeting proteins to the cell division site. We show that loss of *dolP* affects OM fluid-ity, which perturbs the BAM complex, suggesting an indirect role for DolP in OMP biogenesis. The insights provided here not only advance our understanding of how DolP functions but provide a basis for beginning to develop drugs to target it.

## Results

### DolP belongs to an extensive family of lipoproteins required for OM homeostasis

In *E. coli*, the *dolP* gene is located downstream of the genes encoding LpoA (an activator of PBP1A) (*Typas et al., 2010*), YraN (a putative Holiday-Junction resolvase), and DiaA (a regulator of chromo-somal replication) (*Ishida et al., 2004*), and two σ$^E$-dependent promoters are found immediately upstream of the *dolP* gene (*Dartigalongue et al., 2001*; *Figure 1A*). Bioinformatic analyses pre-dicted that *dolP* encodes a lipoprotein with two putative domains of unknown function, termed BON domains (*Yeats and Bateman, 2003*), as well as a Lol-dependent OM targeting signal sequence where acylation was predicted to occur on cysteine residue C19. To test the hypothesis that DolP is localised to the periplasmic face of the OM, we raised an antiserum to the protein to probe subcellular fractions. DolP was found in the Triton X-100 insoluble fraction of the *E. coli* cell envelope along with other OM proteins. As a control for the antiserum, DolP was absent from Triton X-100 insoluble fractions of cell envelopes harvested from *E. coli* Δ*dolP* (*Figure 1—figure supple-ment 1A*). Furthermore, expression of a C19A point mutant, preventing N-terminal acylation, effec-tively eliminated DolP from the OM fractions (*Figure 1—figure supplement 1B*). Unlike the lipoproteins BamC and Lpp, which can be surface localized (*Cowles et al., 2011*; *Webb et al., 2012*), DolP was not accessible to antibody or protease in intact *E. coli* cells. However, DolP could be labelled and degraded when OM integrity was compromised (*Figure 1—figure supplement 1C, D*), confirming that DolP is predominantly targeted to the inner leaflet of the OM, localizing it within the periplasmic space.

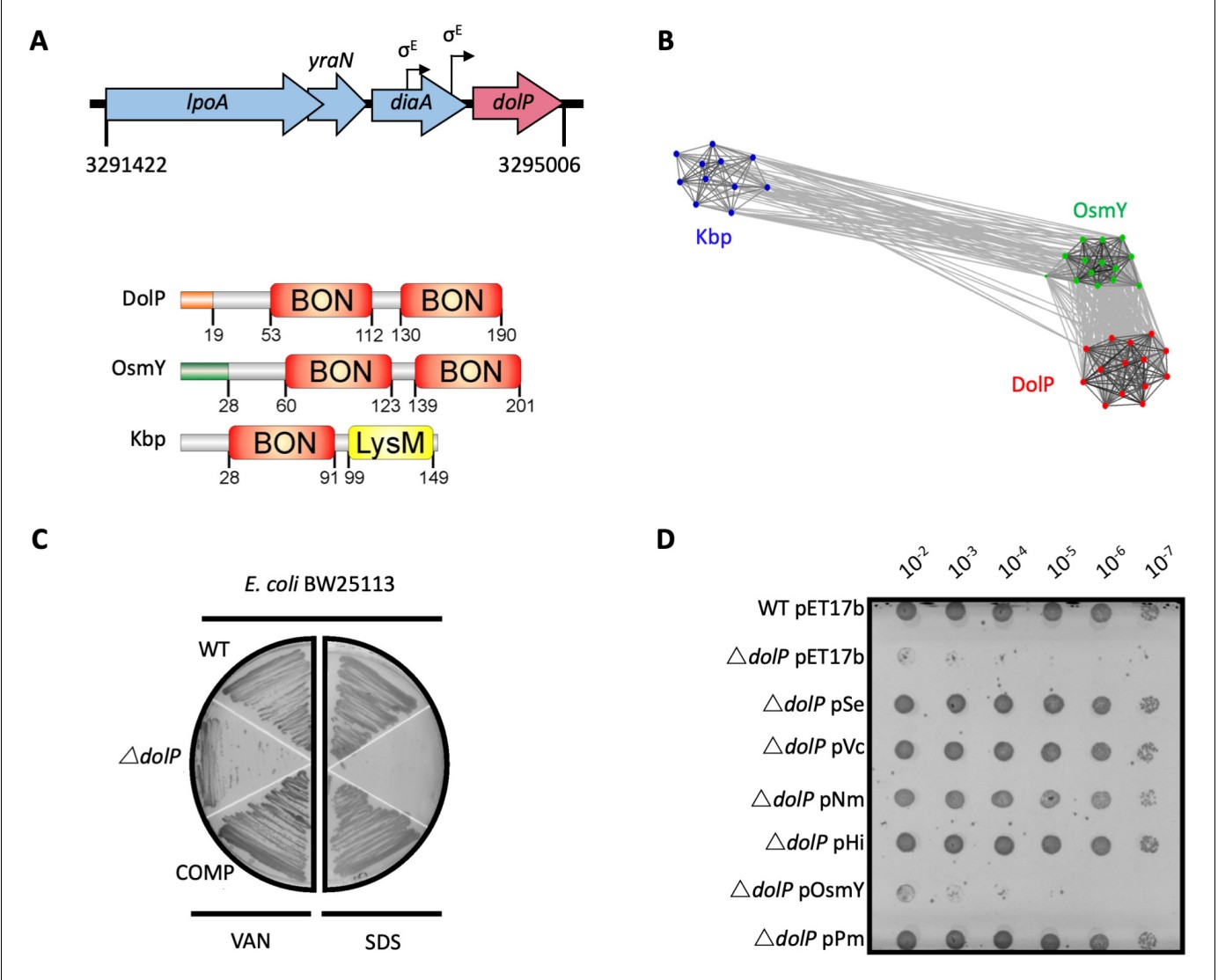

**Figure 1.** DolP is a conserved BON-domain protein with a distinct role in OM homeostasis. (A) In *E. coli*, *dolP* is located downstream of *diaA* and encodes a lipoprotein with a signal sequence (orange) and two BON domains (red). The signal sequence is cleaved by LspA, the cysteine at position 19 acylated by Lgt and Lnt and finally the protein is targeted to the OM by the Lol system (*Figure 1—figure supplement 1*). *E. coli* contains three BON-domain proteins. DolP shares a similar domain organisation with OsmY, which encodes a periplasmic protein that possesses a signal sequence (green) which is recognised and cleaved by the signal peptidase LepB. Kbp is more divergent from DolP and OsmY, has no predictable signal sequence and is composed of BON and LysM domains (*Figure 1—figure supplement 2*). (B) DolP, OsmY and Kbp are widespread amongst proteobacteria, and cluster into three distinct groups based on the program CLANS (*Frickey and Lupas, 2004*) with connections shown for a P value cut-off of $<10^{-2}$ (*Table 4*). (C) Growth phenotypes for mutant isolates lacking DolP (Δ*dolP*), wild-type strain (WT) or the complemented mutant (COMP). Strains were grown on LB agar containing vancomycin (100 μg/ml) or sodium dodecyl sulphate (SDS; 4.8%). (D) DolP from diverse proteobacterial species expressed in an *E. coli* Δ*dolP* strain restores growth in the presence of vancomycin as assessed by a serial dilution plate growth assay. Plasmids expressing OsmY do not complement the defect.

The online version of this article includes the following source data and figure supplement(s) for figure 1:

**Figure supplement 1.** DolP is an OM lipoprotein.
**Figure supplement 1—source data 1.** Subcellular localisation of DolP.
**Figure supplement 2.** BON domain (Pfam: PF04972) containing proteins.
**Figure supplement 3.** DolP has a distinct function from OsmY and Kbp.
**Figure supplement 4.** Phenotypes of *E. coli* BW25113 Δ*dolP*.
**Figure supplement 4—source data 1.** Comparison of bacterial growth rates of wild type and yraP mutant.
**Figure supplement 5.** Localisation of DolP to the OM is required for function.
**Figure supplement 5—source data 1.** The influence of signal sequences on DolP localisation.

**Table 1.** Taxonomic distribution of BON family domain architectures.

| Cluster number* | Total number of UniRef100† | proteins† | Major domain architecture in cluster§ | α | β | γ | δ | ε | ζ | Aci†† | Act†† | Bac†† | Chl†† | Chl†† | Chl†† | Cya†† | Dei†† | Fib†† | Fir†† | Gem†† | Nit†† | Pla†† | Spi†† | Syn†† | The†† | The†† | The†† | Ver†† |
|---|---|---|---|---|---|---|---|---|---|---|---|---|---|---|---|---|---|---|---|---|---|---|---|---|---|---|---|---|
| 1 | 1280 | 2723 | OsmY-like and 1 x BON | 41 (89)¶,** | 176 (533) | 1484 (1830) | 33 (56) | 12 (12) | 1 (1) | 6 (12) | 2 (3) | 5 (5) | 3 (11) | | 3 (4) | 43 (65) | 1 (1) | | 13 (13) | 1 (2) | 1 (1) | 14 (30) | 9 (9) | | | 1 (1) | 1 (1) | 7 (19) |
| 2 | 833 | 2395 | DolP-like | 97 (103) | 330 (335) | 1892 (1919) | 15 (17) | 2 (2) | | | | | | | | | | | 1 (1) | 1 (2) | | 1 (2) | | | | 1 (1) | | |
| 3 | 579 | 690 | three x BON + 1 x BON | 95 (187) | 108 (255) | 35 (36) | 18 (28) | | | 7 (23) | 14 (25) | 14 (30) | 3 (21) | 2 (2) | | 6 (10) | 5 (7) | 1 (1) | 32 (32) | 1 (2) | | 12 (27) | | | 1 (1) | | | |
| 4 | 476 | 537 | BON + secretin | 207 (276) | 77 (80) | 70 (117) | 32 (34) | | | 4 (4) | 1 (1) | | | | 3 (3) | | | | 10 (11) | | 1 (1) | 7 (7) | | 1 (1) | | | | |
| 5 | 409 | 1570 | Kbp-like | 66 (66) | 131 (132) | 1323 (1328) | 1 (1) | 1 (1) | | | | 31 (31) | | | | | 5 (5) | | 1 (1) | | | | | 1 (1) | | | | |
| 6 | 282 | 300 | CBS + CBS + BON | 82 (136) | 17 (29) | 4 (4) | | | | | 53 (127) | 4 (4) | | | | | | | | | | | | | | | | |
| 7 | 220 | 318 | BON + BON + OmpA | 157 (161) | 55 (57) | 9 (11) | | | | | 62 (64) | 1 (1) | | | | 19 (23) | | | | | 1 (1) | | | | | | | |
| 8 | 70 | 75 | BON + Mschannel | 31 (32) | 1 (1) | 24 (25) | 2 (3) | | | | | | | | | | 1 (1) | | | | | 8 (13) | | | | | | |
| 9 | 52 | 52 | one x BON | | 1 (1) | | | | | | | | | | | 42 (51) | | | | | | | | | | | | |
| 10 | 43 | 80 | one x BON and 1 x DUF2204 | | 1 (1) | 1 (1) | | | | | 77 (77) | | | | | | | | | | | | | | | | | 1 (1) |
| 11 | 33 | 87 | 1–2 X Forkhead + BON | 2 (2) | 4 (4) | | | | | | | 2 (2) | | 78 (79) | | | | | | | | | | | | | | |
| 12 | 30 | 33 | one x BON | 26 (27) | | | 3 (3) | | | 1 (1) | 1 (1) | | | | 1 (1) | | | | | | | 1 (1) | | | | | | |
| | | | smaller cluster/unclustered: | | | | | | | | | | | | | | | | | | | | | | | | | |
| | 83 | 109 | | 22 (29) | 19 (19) | 25 (25) | | | | 9 (9) | 9 (9) | | | | 1 (1) | | | | 4 (12) | | 2 (2) | | | | | 1 (1) | |

* The main twelve clusters were analysed, all proteins falling into smaller clusters were summarised into the single category 'smaller cluster'.

†, ‡, §, ¶ Shown are the number of UniRef100 used in the clustering approach†, the corresponding number of proteins derived from the HMMER search‡; the observed major domain architectures§ and the number of unique protein sequences (in brackets)¶ as well as the number of unique organisms mapped to the bacterial (Sub)Phyla**.

†† Acidobacteria, Actinobacteria, Bacteroidetes, Chlamydiae, Chloroflexi, Chlorobi, Cyanobacteria, Deinococcus-Thermus, Fibrobacteres, Firmicutes, Gemmatimonadetes, Nitrospirae, Planctomycetes, Spirochaetes, Synergistetes, Thermobaculum, Thermodesulfobacteria, Thermotogae, Verrucomicrobia.

Further in silico analyses revealed the DolP lipoprotein was conserved across diverse species of Proteobacteria and is present even in organisms with highly-reduced genomes for example *Buchnera* spp (*Table 1* and *Supplementary file 1*). The genome of *E. coli* contains three BON-domain-containing proteins: DolP, OsmY, and Kbp. DolP shares a dual BON-domain architecture and 29.5% sequence identity with OsmY, which is distinguished from DolP by a canonical Sec-dependent signal sequence. In contrast, Kbp consists of single BON and LysM domains and lacks a discernible signal sequence (*Figure 1A*). Our comprehensive analysis found seven predominant domains co-occurring with BON in different modular protein architectures across bacterial phyla, suggesting specialised roles for BON domains (*Table 1* and *Figure 1—figure supplement 2*). Clustering analyses of sequences obtained by HMMER searches revealed DolP, OsmY and Kbp are distributed throughout the α, β, and γ-proteobacteria and form distinct clusters indicating that DolP has a role that is independent of OsmY and Kbp (*Figure 1B*). Our analyses demonstrated that OsmY and Kbp are not functionally redundant with DolP and isogenic mutants show distinct phenotypes, therefore confirming a distinct role for DolP in *E. coli* (*Figure 1—figure supplement 3*).

Previously, we demonstrated that loss of *dolP* in *S. enterica* conferred susceptibility to vancomycin and SDS, suggesting DolP plays an important role in maintaining the integrity of the OM (*Morris et al., 2018*). Further evidence of a role for DolP in maintaining OM integrity is shown by *E. coli ΔdolP* susceptibility to vancomycin, SDS, cholate, and deoxycholate (*Figure 1C* and *Figure 1—figure supplement 4A*). Resistance could be restored by supplying *dolP* in trans (*Figure 1C*). Despite evidence for disrupted OM integrity, the growth rate observed for the *dolP* mutant strain was identical to that of the parent, and scanning-electron microscopy revealed no obvious differences in cell size or shape (*Figure 1—figure supplement 4B,C*). To determine whether DolP is broadly required for OM homeostasis, plasmids expressing DolP homologues from *S. enterica*, *Vibrio cholerae*, *Pasteurella multocida*, *Haemophilus influenza,* and *Neisseria meningitidis* were shown to restore the OM barrier function of the *E. coli ΔdolP* mutant (*Figure 1F*). Finally, either replacement of the DolP signal sequence with that of PelB (*Tsang et al., 2017*), which targets the protein to the periplasmic space, or mutation of the signal sequence to avoid OM targeting *via* the Lol system, prevented complementation of the *ΔdolP* phenotype (*Figure 1—figure supplement 5*). Together these results support a conserved role for DolP in maintenance of OM integrity throughout Gram-negative bacteria and demonstrate that localisation of DolP to the inner leaflet of the OM is essential to mediate this function.

## The structure of DolP reveals a dual BON-domain lipoprotein

To gain further insight into the function of DolP, the structure of full-length mature *E. coli* DolP was determined by NMR spectroscopy. To promote native folding of DolP, the protein was overexpressed in the periplasm using a PelB signal sequence; the N-terminal cysteine was removed to prevent acylation and provide for rapid purification of the soluble protein. Purified DolP was processed, soluble and monomeric, as confirmed by analytical ultra-centrifugation and size exclusion chromatography (*Figure 2—figure supplement 1*). Using a standard Nuclear Overhauser Effect (NOE)-based approach, a convergent ensemble was calculated from the 20 lowest-energy solution structures, revealing two BON domains facing away from each other and offset by ~45° (*Figure 2A* and *Figure 2—figure supplement 2*). The individual BON1 (Residues 45–112) and BON2 (Residues 114–193) domains have C-alpha backbone root mean square deviations (RMSDs) of 0.3 and 0.3 Å, respectively, and an overall global RMSD of 0.5 Å (*Table 2*). Despite having low sequence identity (24.7%) each BON domain consists of a three-stranded mixed parallel/antiparallel β-sheet packed against two α-helices yielding an αββαβ topology. The two BON domains present high structural homology and superpose with an RMSD of 1.8 Å over C-alpha backbone (*Figure 2—figure supplements 2* and *3*). Notably, BON1 is embellished by an additional short α1* helix between BON1:α1 and BON1:β1 (*Figure 2A* and *Figure 2—figure supplements 2* and *3*). The N-terminal acylation site is connected through a 27 amino acid dynamic unstructured linker (*Figure 2B*). The molecular envelope of full-length DolP calculated by small-angle X-ray scattering (SAXS) accommodated the NMR-derived structure of DolP and supported the presence of a flexible N-terminal extension. The experimentally determined scattering curve fit the NMR-derived structure with a χ (*Konovalova et al., 2017*) of 1.263, confirming the accuracy of the NMR-derived structure and an exclusively monomeric state (*Figure 2C* and *Figure 2—figure supplement 4*).

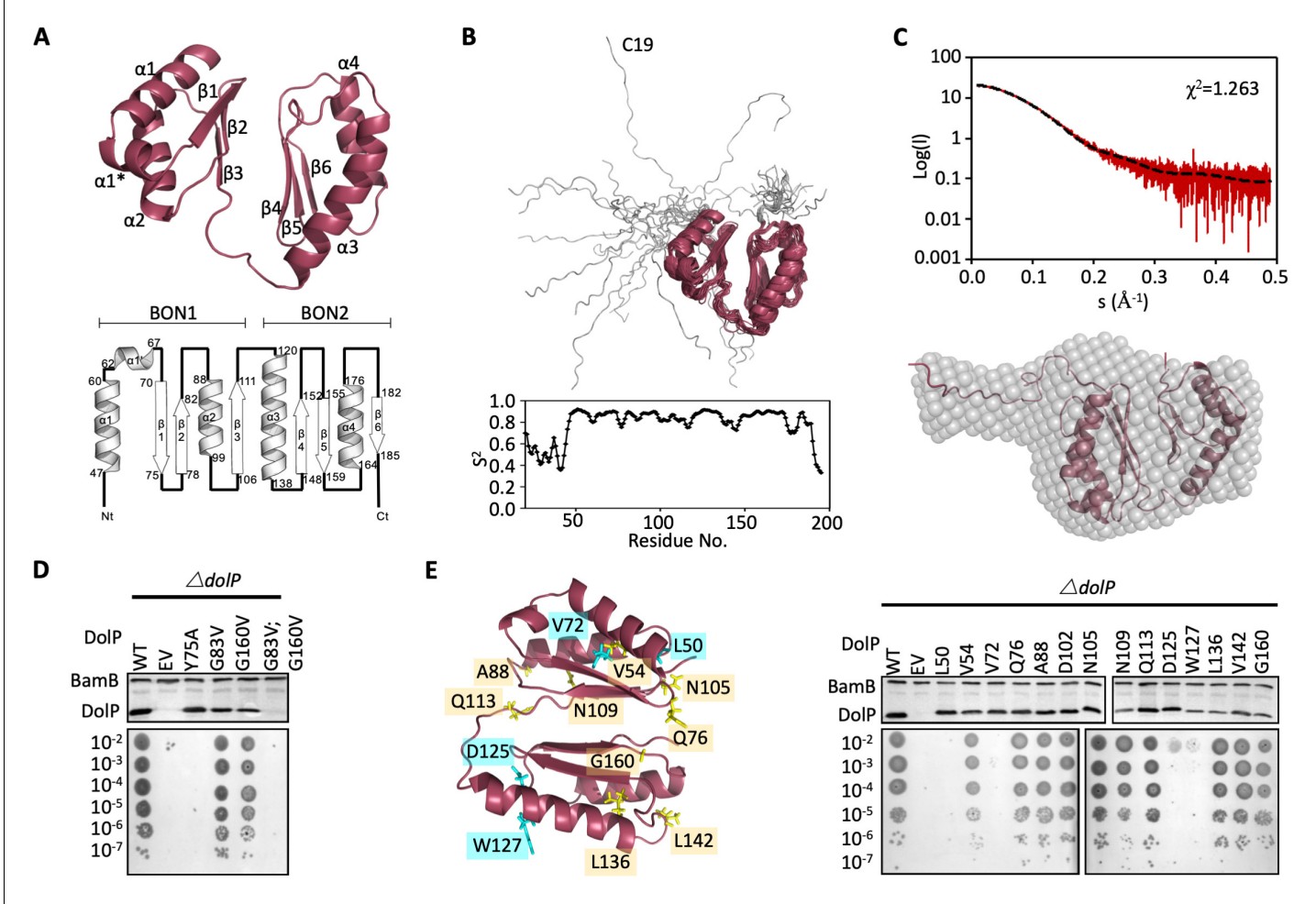

**Figure 2.** Structure of DolP. (**A**) Solution structure and topology of DolP, with α helices, β strands and termini labelled. (**B**) Backbone model of the 20 lowest-energy solution structures of DolP. The core folded domain is highlighted in red whilst the flexible N-terminal is shown in grey. The dynamic nature of the linker was demonstrated from S2 order parameter analysis calculated from chemical shift assignments using TALOS+. (**C**) Small-angle X-ray scattering curve of DolP with corresponding best fit of the solution structure of DolP. Best fit calculated based on the core DolP solution structure with flexibility accommodated in residues 20–46, 112–118, and 189–195. The corresponding *ab-initio* bead model is shown calculated using Dammif (*Franke and Svergun, 2009*) based solely on the scattering data. (**D**) Western blots of total protein extracts show plasmid-mediated expression of DolP in *E. coli* Δ*dolP* after site-directed mutation of conserved residues. The empty vector (EV) control is labelled and WT represents wild-type DolP. The presence of the OM lipoprotein BamB was used as a control. Colony growth assays by serial dilution of mutants on 4.8% SDS reveal which residues are critical for the maintenance of the OM barrier function. (**E**) Structure of DolP showing position of transposon-mediated insertions. Western blots of total protein extracts show plasmid-mediated expression of mutant versions of DolP in *E. coli* Δ*dolP*. The empty vector (EV) control is labelled and WT represents wild-type DolP. Colony growth assays by serial dilution of mutants on 4.8% SDS reveal which insertions abolish DolP function. Blue labels represent position of non-functional insertions. Orange labels represent position of tolerated insertions. The presence of the OM lipoprotein BamB was used as a control.

The online version of this article includes the following source data and figure supplement(s) for figure 2:

**Source data 1.** Influence of site directed mutagenesis of DolP of protein production and stability.

**Source data 2.** S2 order parameter analysis.

**Figure supplement 1.** DolP is monomeric.

**Figure supplement 2.** Structural analysis of the DolP BON domains.

**Figure supplement 3.** Alignment of DolP sequences from diverse proteobacterial species.

**Figure supplement 4.** Additional SAXS analysis of DolP.

**Figure supplement 5.** Representation of DolP interdomain interactions highlighting the location of interdomain NOEs identified.

**Figure supplement 6.** SAXS processing analysis.

**Table 2.** Structural statistics of the ensemble of 20 DolP solution structures.

| | DolP |
|---|---|
| **Completeness of resonance assignments†** | |
| Aromatic completeness | 74.14% |
| Backbone completeness | 98.42% |
| Sidechain completeness | 84.84% |
| Unambiguous CH2 completeness | 100% |
| Unambiguous CH3 completeness | 100% |
| Unambiguous sidechain NH2 completeness | 100% |
| Conformationally restricting restraints‡ | |
| Distance restraints | |
| Total NOEs | 2930 (2762) |
| Intra residue (i = j) | 408 (374) |
| Sequential (\| i − j \|=1) | 869 (783) |
| Medium range (1 < \| i - j \|<5) | 773 (741) |
| Long range (\| i − j \|≥5) | 880 (866) |
| Interdomain | 38 |
| Dihedral angle restraints | 258 |
| Hydrogen bond restraints | 128 |
| No. of restraints per residue | 16.6 (20.9) |
| No. of long range restraints per residue | 5.0 (6.5) |
| Residual restraint violations‡ | |
| Average No. of distance violations per structure | |
| 0.2 Å-0.5 Å | 3.55 |
| >0.5 Å | 0 |
| Average No. of dihedral angle violations per structure | |
| >5o | 0 (max 4.8) |
| Model quality‡ | |
| Global (residues 46–190) | |
| Rmsd backbone atoms (Å)§ | 0.5 |
| Rmsd heavy atoms (Å)§ | 0.9 |
| Domain 1 (Residues 46–112) | |
| Rmsd backbone atoms (Å) | 0.3 |
| Rmsd heavy atoms (Å) | 0.7 |
| Domain 2 (Residues 118–190) | |
| Rmsd backbone atoms (Å) | 0.3 |
| Rmsd heavy atoms (Å) | 0.8 |
| Rmsd bond lengths (Å) | 0.005 |
| Rmsd bond angles (o) | 0.6 |
| MolProbity Ramachandran statistics‡.§ | |
| Most favoured regions (%) | 95.1 |
| Allowed regions (%) | 4.3 |
| Disallowed regions (%) | 0.7 |
| Global quality scores (raw/Z score)‡ | |
| Verify 3D | 0.38 /- 1.28 |
| Prosall | 0.52 /- 0.54 |

*Table 2 continued on next page*

*Table 2 continued*

| | DolP |
|---|---|
| **Completeness of resonance assignments†** | |
| Procheck (phi-psi)[d] | −0.28 /- 0.79 |
| Procheck (all)[d] | −0.75 /- 4.44 |
| Molprobity clash score | 47.99 /- 6.71 |
| Model Contents | |
| Ordered residue ranges§ | 45–193 |
| Total number of residues | 178 |
| BMRB accession number | 19760 |
| PDB ID code | 7A2D |

\* Structural statistics computed for the ensemble of 20 deposited structures.

† Computed using AVS software (**Moseley et al., 2004**) from the expected number of resonances, excluding highly exchangeable protons (N-terminal, Lys, amino and Arg guanido groups, hydroxyls of Ser, Thr, and Tyr), carboxyls of Asp and Glu, non-protonated aromatic carbons, and the C-terminal His$_6$ tag.

‡ Calculated using PSVS version 1.5 (**Bhattacharya et al., 2007**). Average distance violations were calculated using the sum over r$^{-6}$.

§ Based on ordered residue ranges [S(φ) + S(ψ)>1.8].

Values in (brackets) refer to the core structured region.

The two BON domains pack against each other *via* their β-sheets through contacts mediated directly by Y75 and V82 in BON1 and T150, G160, L161 and T188 in BON2 with a total of 38 inter-domain NOEs (**Figure 2D**, **Figure 2—figure supplement 5**, **Table 3**). This interdomain orientation is consistent with SAXS analysis (**Figure 2C**) and appears to be essential for function as the mutation Y75A abolishes function (**Figure 2D**). Single point mutations (G83V and G160V) of the highly conserved glycine residues had less effect, however the double mutant was non-functional (**Figure 2D** and **Figure 2—figure supplement 3**). Since the latter protein was not detectable by Western immunoblotting this is likely due to structural instability (**Figure 2D**).

The elements of DolP that are required for function were mapped using an unbiased linker-scanning mutagenesis screen. The resulting DolP derivatives, containing in-frame 5-amino-acid insertions, were tested for stability by Western immunoblotting. Functional viability was assessed by their capacity to restore growth of *E. coli* Δ*dolP* in the presence of SDS (**Figure 2E**). Seven mutants occurred in the signal sequence and the linker region and were not considered further. Eight insertions were identified in BON1, with insertions at positions L50 (BON1:α1) and V72 (BON1:β1) failing to complement the Δ*dolP* defect whereas the rest were well tolerated. Five insertions were found in BON2, with those at positions L136, L142, and G160 being well tolerated. The remaining insertions at positions D125 and W127 occurred in BON2:α1 but failed to complement the Δ*dolP* phenotype. None of these mutations abolished protein expression. These data indicate the importance of BON2:α1 in maintaining DolP function and OM integrity (**Figure 2E**).

## DolP binds specifically to anionic phospholipids via BON2

Given that OM permeability defects are often associated with the loss or modification of molecular partners, we sought to identify DolP ligands. Scrutiny of the literature revealed high-throughput protein:protein interaction data (**Carlson et al., 2019**; **Babu et al., 2018**) indicating that DolP co-located with components of the BAM complex in the OM. As the loss of multiple genes encoding different components of a single pathway can have additive phenotypes, such as decreased fitness, we investigated strains with dual mutations in *dolP* and genes coding the non-essential BAM complex components *bamB* or *bamE*. We observed that simultaneous deletion of *dolP* and *bamB* or *bamE* lead to negative genetic interactions and increased rates of cell lysis (**Figure 3—figure supplement 1A,B**), suggesting a potential interaction. However, despite these genetic interactions, in our hands no significant interaction could be detected between DolP and the BAM complex through immunoprecipitations (**Figure 3—figure supplement 1C**) and no significant change in overall OMP

**Table 3.** Interdomain NOE restraints identified by Cyana during automated NOE assignment and structure calculation.

| Proton pair | Intensity | Distance (Å) |
|---|---|---|
| TYR 75 HD1 - THR 188 HA | Weak | 5.5 |
| TYR 75 HE1 - GLY 160 HA2 | Weak | 5.4 |
| TYR 108 HE1 - ALA 186 HA | Weak | 5.5 |
| TYR 108 HE2 - ALA 186 HA | Weak | 5.5 |
| TYR 108 HE1 - ALA 186 HB | Weak | 5.1 |
| TYR 75 HD1 - ALA 186 HB | Weak | 5.2 |
| TYR 75 HE1 - LEU 161 HA | Weak | 5.2 |
| TYR 75 HE1 - LEU 161 HB3 | Weak | 5.4 |
| TYR 75 HE1 - LEU 161 HG | Weak | 5.5 |
| TYR 75 HE1 - LEU 161 HD1 | Weak | 4.9 |
| TYR 75 HE1 - LEU 161 HD2 | Weak | 4.9 |
| THR 73 HG2 - ALA 186 HB | Weak | 5.5 |
| LYS 78 HD2 - PHE 187 hr | Weak | 5.5 |
| LYS 78 HD3 - PHE 187 hr | Weak | 5.5 |
| TYR 75 HD1 - HET 159 HA | Weak | 5.5 |
| TYR 108 HD1 - ALA 186 HB | Weak | 5.5 |
| GLN 76 HE22 - LEU 161 HB2 | Weak | 5.2 |
| GLN 76 HE22 - LEU 161 HG | Weak | 5.1 |
| GLN 76 HE22 - LEU 161 HD1 | Weak | 4.5 |
| GLN 76 HE22 - LEU 161 HD2 | Weak | 4.5 |
| TYR 75 HD1 - THR 188 HG2 | Weak | 4.2 |
| TYR 75 HE1 - LEU 161 hr | Weak | 4.3 |
| TYR 75 HE1 - VAL 162 hr | Weak | 5.5 |
| TYR 75 HE1 - LEU 161 HB2 | Weak | 4.1 |
| TYR 75 HE1 - THR 188 HG2 | Weak | 4.1 |
| TYR 75 HE1 - THR 188 hr | Weak | 5.5 |
| TYR 75 HE1 - GLY 160 hr | Weak | 4.8 |
| TYR 75 HD1 - GLY 160 hr | Weak | 4.7 |
| THR 73 HG2 - HET 159 HG | Weak | 4.4 |
| TYR 75 HE1 - LEU 161 HD | Weak | 4.0 |
| TYR 75 HE2 - LEU 161 HD | Weak | 5.1 |
| GLN 76 HE21 - LEU 161 HD | Medium | 3.7 |
| GLN 76 HE22 - LEU 161 HD | Medium | 3.7 |
| LYS 78 HG - PHE 187 hr | Weak | 4.9 |
| LYS 78 HD - ALA 186 HB | Weak | 5.1 |
| LYS 78 HD - PHE 187 hr | Weak | 4.7 |
| LYS 78 HE - PHE 187 hr | Weak | 5.3 |
| ARG 112 HA - ARG 182 HB | Weak | 5.3 |

levels was observed (*Supplementary file 2* and *Figure 3—figure supplement 1D*). Analyses of purified OM fractions revealed no apparent differences in LPS profiles (*Figure 3—figure supplement 2A*), or phospholipid content (*Figure 3—figure supplement 2B*) between the parent and the *dolP* mutant. No significant increase in hepta-acylated Lipid A was observed in the absence of DolP, indicating that the permeability defect is also not due to loss of OM lipid asymmetry (*Figure 3—figure*

supplement 2C). In contrast, Δ*dolP* cells were found to have an increase in membrane fluidity (*Figure 3—figure supplement 2D*) as assessed by staining with the membrane intercalating dye pyrenedecanoic acid (PDA), which undergoes a fluorescence shift upon formation of the excimer, an event which is directly related to membrane fluidity (*Storek et al., 2019*). Considering that *bamB* mutants are sensitive to increased membrane fluidity (*Storek et al., 2019*), these data suggest that the genetic interaction between *dolP* and *bamE* or *bamB*, observed here, is facilitated indirectly through changes to membrane fluidity on the loss of DolP.

The *dolP* mutant has changed to membrane fluidity and that BON domains are suggested to bind phospholipids (*Yeats and Bateman, 2003*), therefore we sought to test whether DolP interacts with phospholipids. A set of potential ligands were screened by chemical shift perturbation (CSP) analysis, including *E. coli* OM lipids embedded in micelles. DolP bound specifically to micelles containing the anionic phospholipids phosphatidylglycerol (PG) and cardiolipin (CL) but not to micelles devoid of PG or CL, or those containing the zwitterionic phospholipid phosphatidylethanolamine (PE) (*Figure 3A*, *Figure 3—figure supplement 3*, *Figure 4A*). Significant CSPs were noted for A74, G120-I128, K131-R133, Q135-L137, V142-S145, I173, and S178-V180. The perturbed residues were mapped to the structure, revealing a single extensive binding site centred on BON2:α1 that was sufficiently large to contact several lipid molecules (*Figure 3A*). A dissociation constant ($K_d$) of ~100 mM (monomeric DHPG) was measured (*Figure 3—figure supplement 4*). No lipid interaction was seen for any BON1 domain residue, emphasising the specialised role of BON2, which not only differs from DolP BON1, but also from the BON domains of OsmY and Kbp (*Figure 2—figure supplement 3*). Analysis of the electrostatic surface reveals a large negative surface potential on BON1:α1, which is absent in BON2:α1 and may act to repel BON1 from PG, whilst BON2:α1 uniquely harbours an aromatic residue W127 in the observed PG- binding site (*Figure 4—figure supplement 1*).

As the BON2 domain contained a particularly large PG-specific interaction site, we sought to resolve the micelle-complexed structure of mature DolP. Intermolecular structural restraints were obtained from paramagnetic relaxation enhancements (PRE) obtained by incorporating 5-doxyl spinlabelled phosphatidyl choline (PC) and 1,2-dimyristoyl-*sn*-glycero-3-phospho-(1′-rac-glycerol) (DMPG) into a *n*-dodecylphosphocholine (DPC) micelle and by measuring CSPs. The complexed structure was calculated using HADDOCK (*Dominguez et al., 2003*) with 18 PRE distance restraints and side chains of the 25 chemical shift perturbations, with final refinement in water (*Figure 3B*). The amino acids G120-T130 and V132-S139 were observed to insert into the micelle interior based on the PRE and CSP data. This reveals an unprecedented burial of the BON2:α1 helix, which spans the entirety of the L119-S139 sequence. The protein-micelle interface buries $1358 \pm 316$ $Å^2$ and to our knowledge represents the most extensive structured surface of a membrane:protein interface resolved to date. The surface forms intimate contacts with at least six proximal phospholipid headgroups through an extensive network of highly populated hydrogen bonds and electrostatic interactions. Whilst the side chains of residues G120, S123, W127, T130, and S134 intercalate between the acyl chains, E121, N124, T126, I128, K131, R133, and Q135 buttress the interface (*Figure 3B*). This element was also functionally important based on our transposon screen (*Figure 2E*), and was further confirmed as being essential by directed mutagenesis. Mutations within the PG-binding BON2:α1 disrupt the function of DolP, the most critical of which are W127E and L137E; W127 is located in the centre of the binding site that penetrates deep into the core of the PG micelle, and L137 is located at the periphery of the helix (*Figure 3B*, *Figure 4B* and *Figure 4—figure supplement 2*). Not only does mutation of W127 lead to loss of function, but introduction of the W127E mutation was shown to abolish binding of DolP to PG micelles as observed by a loss of CSPs within BON2:α1 (*Figure 4C*). Notably, the BON2:α1 structure presents an extended α-helix when compared to BON1:α1 (*Figure 2—figure supplements 2* and *3*). The helical extension in BON2:α1 contains the W127 anionic phospholipid-binding determinant of DolP. This further implicates W127, which is absent in BON1 and OsmY, in specialisation of DolP BON2 for phospholipid binding.

## Phospholipid-binding guides DolP localisation to the cell division site

DolP binds anionic phospholipid, which demonstrates sub-cellular localisation to sites of higher membrane curvature including the cell poles and division site (*Oliver et al., 2014*; *Renner and Weibel, 2011*; *Mileykovskaya and Dowhan, 2000*). To determine if DolP also shows a preference for such sites, we constructed a plasmid expressing a DolP-mCherry fusion and utilising fluorescence microscopy we observed DolP localised specifically to the cell division site (*Figure 5A*). Considering

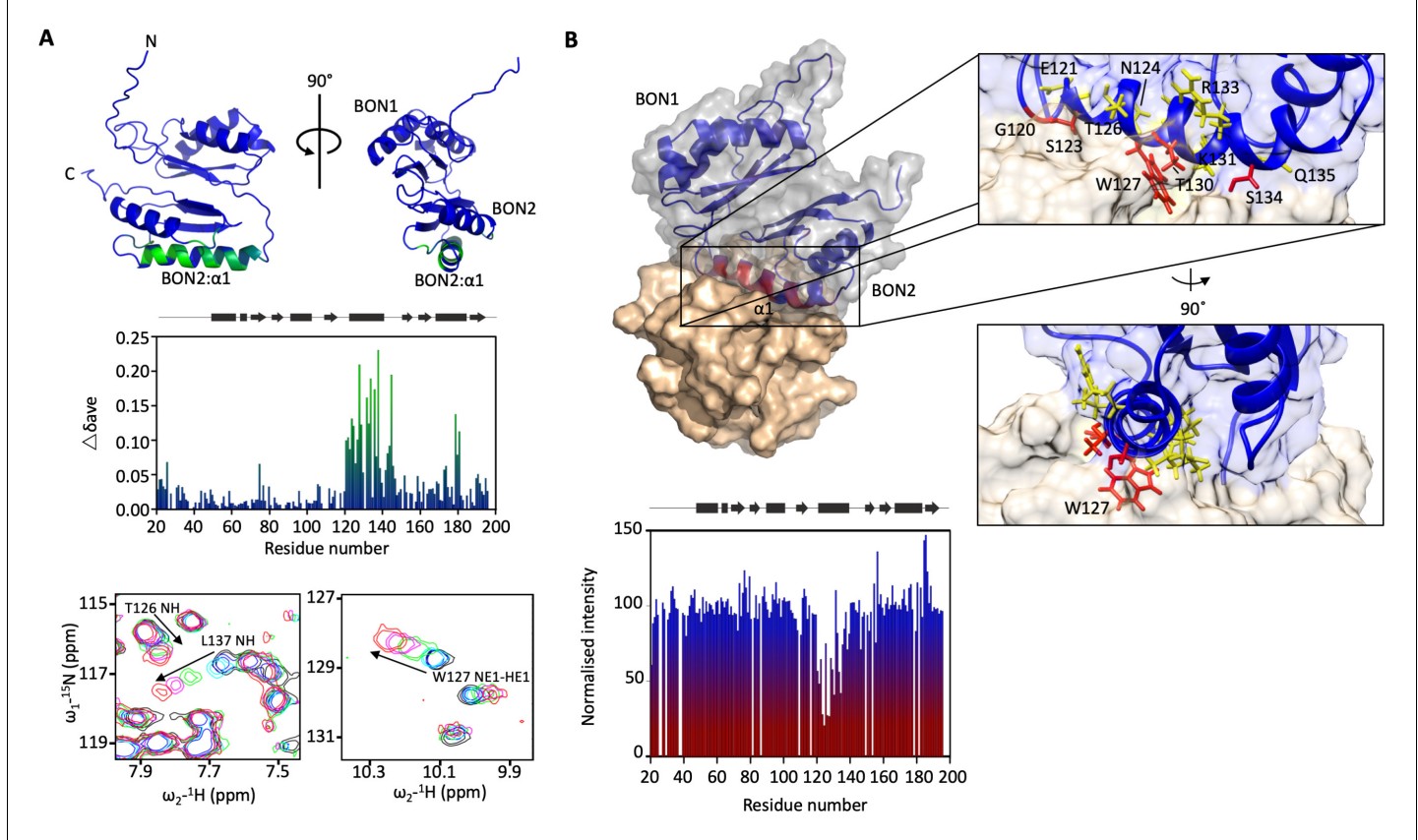

**Figure 3.** DolP BON2:α1 binds phospholipid. (**A**) DolP ribbon structure highlighting residues exhibiting substantial CSPs ($\Delta\delta_{ave}$) upon DHPG micelle interaction. The histogram shows the normalised perturbations induced in each residue's amide signal when DHPG (40 mM) was added to DolP (300 µM). Examples of significant CSPs are shown. (**B**) Histogram showing intensity reductions of $H_N$ signals of DolP induced by adding 5-doxyl PC and DMPG into DPC/CHAPs micelles and the corresponding structure of a representative DolP-micelle complex calculated using CSPs and doxyl restraints using the program HADDOCK. Only the BON2:α1 helix is observed making contact with the micelle surface. No corresponding interaction of the BON1:α1 helix is observed. Zoom panels show burial of BON2:α1 into the micelle. The side chains of DolP residues that intercalate between the acyl chains (G120, S123, W127, T130, and S134) are coloured red. The side chains of residues that buttress the interface (E121, N124, T126, I128, K131, R133, and Q135) are coloured yellow. DolP is shown in blue and the phospholipid micelle is shown in tan.

The online version of this article includes the following source data and figure supplement(s) for figure 3:

**Source data 1.** Chemical shift perturbations for lipid titration results.

**Source data 2.** Data for HADDOCK calculations of micelle-DolP interactions.

**Figure supplement 1.** *dolP* has genetic interactions with *bamB* and *bamE* but no detectable physical interaction.

**Figure supplement 1—source data 1.** Genetic interactions with DolP.

**Figure supplement 2.** Loss of DolP affects membrane fluidity, but does not affect membrane lipid profiles.

**Figure supplement 2—source data 1.** LPS production in a dolP negative background.

**Figure supplement 2—source data 2.** Phospholipid content of membranes isolated from a dolP mutant.

**Figure supplement 2—source data 3.** Comparison of hepta- and hexa-acylated LPS levels.

**Figure supplement 2—source data 4.** Raw data for membrane fluidity assay.

**Figure supplement 3.** DolP phosphatidylglycerol binding HSQC spectra.

**Figure supplement 4.** Kd estimation from HSQC titration data.

that DolP is non-functional when targeted to the IM (*Figure 1—figure supplement 5*), we investigated if DolP could still localise to the site of cell division when it was mistargeted to the IM; no septal localisation was observed (*Figure 1—figure supplement 5*). Next, we tested whether the phospholipid-binding activity is also required for division site localisation of DolP. We found that introduction of the W127E mutation, which prevents interaction of DolP with PG/CL micelles, abolished division site localisation of DolP (*Figure 5A*). Considering that W127E not only abolished PG/CL binding, but also division site localisation, we concluded that division site localisation of DolP was

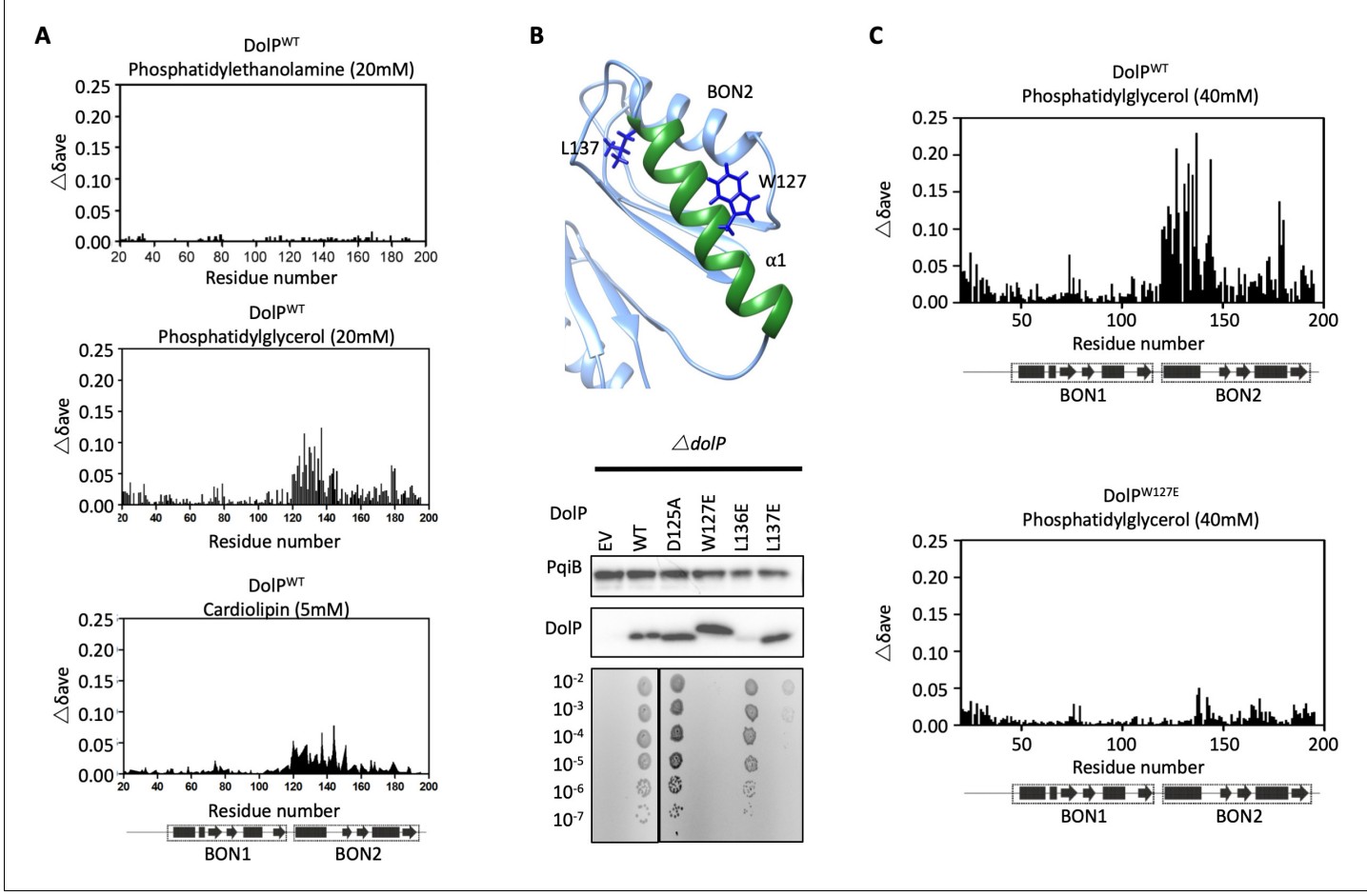

**Figure 4.** DolP specifically recognises anionic phospholipid via BON2:α1. (**A**) Histograms showing the normalised CSP values observed in [15]N-labelled DolP (300 μM) amide signals in the presence of 20 mM 1,2,-dihexanoyl-sn-glycero-3-phosphethanolamine, 20 mM 1,2-dihexanoyl-sn-glycero-3-phospho-(1′-rac-glycerol) and 5 mM cardiolipin.( **B**) Mutagenesis of the BON2:α1 helix residues identified by CSPs. The positions of W127 and L137 are indicated as sticks. Western blots of total protein extracts show plasmid-mediated expression of DolP in *E. coli ΔdolP* after site-directed mutation of amino acid residues. The empty vector (EV) control is labelled and WT represents wild-type DolP. Colony growth assays of *E. coli ΔdolP* complemented with DolP mutants reveal which residues are critical for the maintenance of OM barrier function. The presence of the protein PqiB was used as a control. (**C**) Histograms showing the normalised CSP values observed in [15]N-labelled DolP[WT] or DolP[W127E] mutant (300 μM) amide signals in the presence of 40 mM 1,2-dihexanoyl-sn-glycero-3-phospho-(1′-rac-glycerol).

The online version of this article includes the following source data and figure supplement(s) for figure 4:

**Source data 1.** Effect of site-directed mutations on DolP function.
**Figure supplement 1.** Electrostatic analysis of DolP.
**Figure supplement 2.** Analysis of DolP mutants.

dependent upon binding of DolP to anionic phospholipids, which have previously been shown to be enriched at the division site (*Renner and Weibel, 2011*; *Mileykovskaya and Dowhan, 2000*).

To confirm this result we analysed DolP localisation in a strain that lacks all three cardiolipin synthases and is defective for cardiolipin synthesis, which was confirmed by phospholipid extraction and thin layer chromatography (*Figure 5B*). We observed that DolP localisation is perturbed in the CL⁻ strain, with less dividing cells showing localisation of DolP to the septum (*Figure 5C*). These effects are further exacerbated in a strain that does not synthesise the major cell anionic phospholipids phosphatidylglycerol or cardiolipin, as confirmed by phospholipid extraction and thin layer chromatography (*Figure 5B*). Loss of both phosphatidylglycerol and cardiolipin synthesis worsened the severity of the localisation defect with less septal localisation and a significant proportion of cells showing mislocalisation of DolP to patches at the cell poles (*Figure 5C*). Taken together these data demonstrate that DolP localisation to the division site is dependent upon interaction with anionic

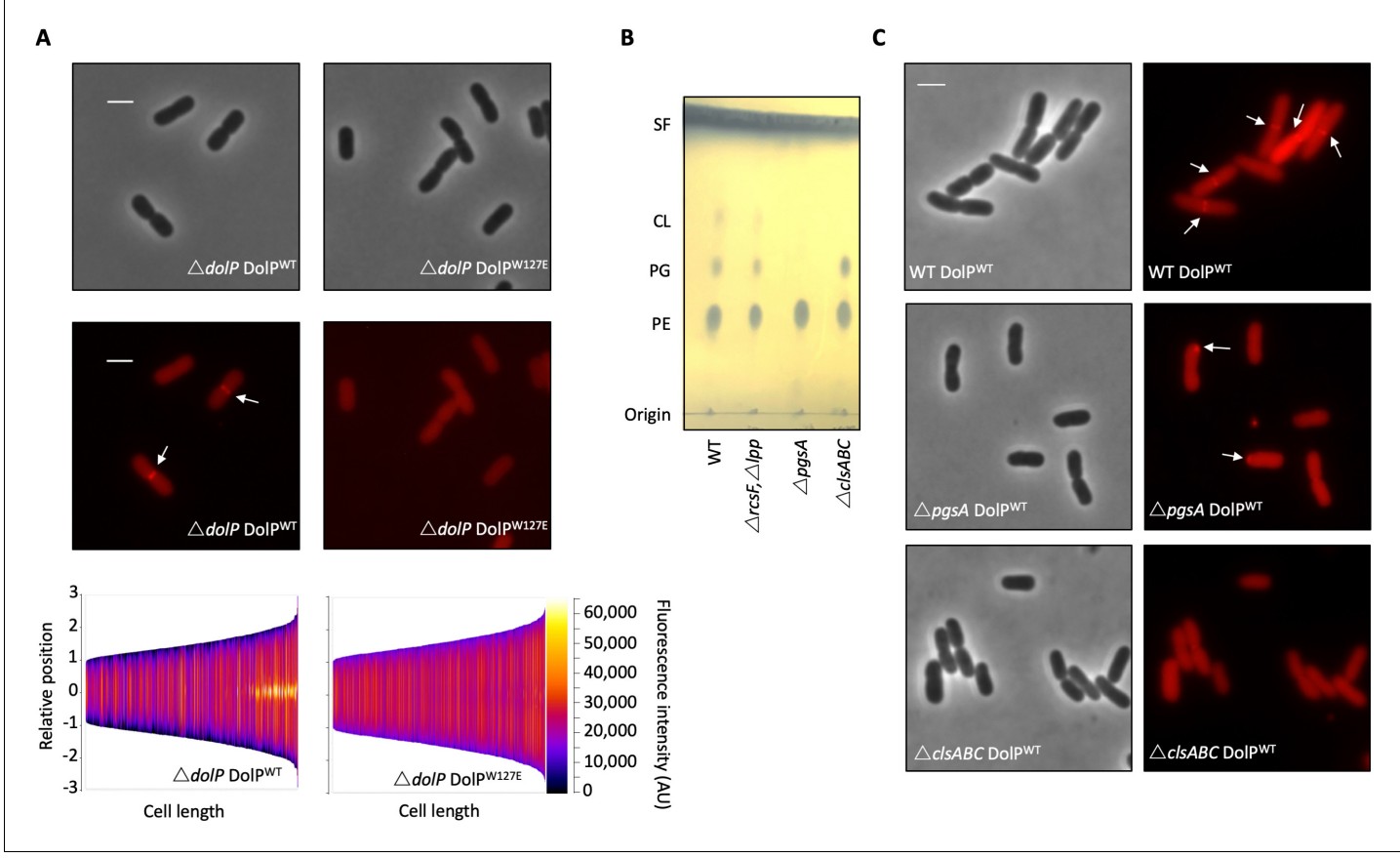

**Figure 5.** Phospholipid binding is required for DolP recruitment to division sites. (**A**) Fluorescence microscopy of ΔdolP cells expressing either DolP[WT]::mCherry or DolP[W127E]::mCherry from the pET17b plasmid after growth to mid-exponential phase (OD$_{600}$ ~0.4–0.8). Scale bars represent 2 μM and both phase contrast and the mCherry channel are shown in greyscale and red respectively. White arrows highlight division site localisation of DolP[WT]-mCherry. Demographic representations of the DolP[WT]-mCherry or DolP[W127E]-mCherry fluorescence intensities measure along the medial axis of the cells. Images of >500 cells were analysed using the MicrobeJ software and sorted according to length where the y-axis represents relative cellular position with 0 being mid-cell and 3 or −3 being the cell poles (*Ducret et al., 2016*). (**B**) Thin layer chromatography of phospholipids extracted from either *E. coli* BW25113 (WT), Δ*rcsF*Δ*lpp*, Δ*rcsF*Δ*lpp*Δ*pgsA* (referred to as Δ*pgsA*) or Δ*clsA*Δ*clsB*Δ*clsC* (referred to as Δcls*ABC*) strains. The *rcsF* and *lpp* genes must be removed in order to prevent toxic build-up of Lpp on the IM in the *pgsA* mutant. Phospholipids were separated using chloroform:methanol:acetic acid (65:25:10) as the mobile phase before staining with phophomolybdic acid and charring.( **C**) Fluorescence microscopy of Δ*pgsA* or Δcls*ABC* cells expressing DolP[WT]mCherry from the pET17b plasmid after growth to mid-exponential phase (OD$_{600}$ ~0.4–0.8). White arrows highlight DolP-mCherry mislocalisation.

The online version of this article includes the following source data for figure 5:

**Source data 1.** Effect of DolP-anionic phospholipid interactions on DolP localisation.

phospholipid *via* BON2:α1, and that this interaction and the sub-cellular localisation are required for DolP function.

## Discussion

We have revealed the first structure of a dual-BON-domain protein, a protein architecture that is widely conserved amongst bacteria and therefore provides insight into a diverse range of proteins acting in different organisms. We also report the first evidence for direct binding of lipids by BON domains. We show that DolP BON2 demonstrates specificity for the anionic phospholipids PG and CL, which have previously been shown to localise to sites of higher membrane curvature including the cell poles and division site (*Oliver et al., 2014*; *Renner and Weibel, 2011*; *Mileykovskaya and Dowhan, 2000*). Interestingly, we detected no phospholipid binding for DolP BON1, which lacks the key W127 phospholipid interaction residue. This key residue is also lacking in the other periplasmic

BON-domain-containing protein in *E. coli*, OsmY. Thus, we have demonstrated a specialised role for DolP in the cell and our data suggests BON domains are not generalist phospholipid-binding domains, as was suggested previously (*Yeats and Bateman, 2003*).

Here, we show for the first time that localisation of DolP to the cell division site is dependent upon recognition of anionic phospholipids by DolP BON2. To our knowledge, this is the only example of this mechanism of localisation to the bacterial division site (*Laloux and Jacobs-Wagner, 2014*). Considering anionic phospholipids also accumulate at the old pole, the question of how DolP specifically recognises the division site remains. We hypothesise that DolP prefers the site of higher positive (convex) curvature found only at the inner leaflet of the OM cell division site in vivo and in the PG micelles used in this study. Previous evidence has shown that inhibition of cell constriction, by the addition of cephalexin, also prevents DolP localisation to future division sites (*Tsang et al., 2017*). This indicates that DolP may require cell constriction for localisation to the division site, therefore lending support to the hypothesis that DolP may recognise membrane curvature. An alternative explanation is that the phospholipid-binding mode of DolP may trigger interaction with some as yet unidentified division site localised protein partner, but no obvious candidates are offered by published envelope interactome data (*Carlson et al., 2019*; *Babu et al., 2018*). Nevertheless, these data reveal that DolP function is dependent on localisation to the division site through phospholipid binding and localisation to the OM through its N-terminal lipid anchor. The model of DolP localisation to the cell division site proposed here also provides some evidence that anionic phospholipids localise to sites of high membrane curvature in the OM. While this has been shown for whole cells (*Oliver et al., 2014*; *Mileykovskaya and Dowhan, 2000*), and the IM through the use of spheroplasts (*Renner and Weibel, 2011*), to our knowledge, no such observation has yet been made for the OM directly. Considering that the OM is significantly different from the IM and is depleted of PG and CL by comparison (*Lugtenberg and Peters, 1976*; *Figure 3—figure supplement 2B*), the localisation of these lipids to sites of negative curvature could be further enhanced by the relative scarcity of these lipids in the OM and this warrants further study.

We have not found a direct mechanism through which DolP maintains OM integrity. No differences in LPS content or OM asymmetry were observed in a *dolP* mutant suggesting DolP does not influence the OM phospholipid recycling Mla pathway or LPS biogenesis. Previous protein:protein interaction studies captured DolP as a near neighbour of two components of the Bam complex, BamD and BamE (*Carlson et al., 2019*; *Babu et al., 2018*). Consistent with this, *dolP* shows synthetic lethality with the gene encoding the periplasmic chaperone SurA, leading to suggestions of a role for DolP in OMP biogenesis (*Onufryk et al., 2005*; *Yan et al., 2019*; *Typas et al., 2008*). However, we were unable to demonstrate a direct interaction between DolP and the BAM complex, and no such interaction has been seen in the extensive studies evaluating the subunit composition and multimeric states of the BAM complex (*Wu et al., 2005*; *Hagan et al., 2010*; *Gunasinghe et al., 2018*; *Knowles et al., 2009*) or in similar studies in *N. meningitidis* (*Bos et al., 2014*). However, while this is in agreement with the fact that DolP is localised to the division site, whereas the Bam complex is uniformly present across the cell surface (*Gunasinghe et al., 2018*), it does not rule out potential transient interactions. Previous observations revealed that the OM is a rigid structure (*Rojas et al., 2018*) that this membrane rigidity stabilises assembly precincts (*Gunasinghe et al., 2018*), and that the activity of the BAM complex is sensitive to increases in membrane fluidity (*Storek et al., 2019*). We suggest that the increased membrane fluidity of the *dolP* cells, demonstrated here, provides a challenging environment for assembly precincts to be maintained. We hypothesise that DolP, perhaps through interactions with peptidoglycan amidases (*Tsang et al., 2017*), might also modulate peptidoglycan remodeling in such a way as to minimise the clash between the periplasmic components of the assembly precinct and the cell wall, which might be exacerbated in regions of high membrane curvature.

In conclusion, this study reports for the first time the direct binding of lipid by BON domains and a new mechanism of protein division site localisation. The indirect link between DolP and the general machinery responsible for outer-membrane biogenesis adds to the recently described role of DolP in the regulation of cell wall amidases during division, therefore potentially placing DolP at the interface between envelope biogenesis processes (*Tsang et al., 2017*). The demonstration that loss of DolP increases sensitivity to antibiotics and membrane disrupting agents, in addition to the decrease in virulence in vivo, and an increase of the efficacy of the *N. meningitidis* vaccine, suggests DolP will provide a useful starting platform for antimicrobial design based on the disruption to regulation of

multiple envelope biogenesis mechanisms (*Morris et al., 2018*; *Giuliani et al., 2006*; *Pizza et al., 2000*).

# Materials and methods

## Key resources table

| Reagent type (species) or resource | Designation | Source or reference | Identifiers | Additional information |
|---|---|---|---|---|
| Strain, strain background (*Escherichia coli*) | BL21(DE3) | Invitrogen | | T7 express, protein expression strain |
| Strain, strain background (*Escherichia coli*) | BW25113 | *Datsenko and Wanner, 2000* | | rrnB3 ΔlacZ4787 ΔphoBR580 hsdR514 Δ(araBAD)567 Δ(rhaBAD)568 galU95 ΔendA9::FRT ΔuidA3::pir (wt) recA1 rph-1 |
| Strain, strain background (*Escherichia coli*) | BW25113 △dolP | This paper | | BW25113 with dolP deleted |
| Strain, strain background (*Escherichia coli*) | BW25113 △lpp,△rcsF | This paper | | BW25113 with lpp and rcsF deleted |
| Strain, strain background (*Escherichia coli*) | BW25113 △lpp,△rcsF,△pgsA | This paper | | BW25113 with lpp, rcsF and pgsA genes deleted |
| Strain, strain background (*Escherichia coli*) | BW25113 △clsA,△clsB,△clsC | This paper | | BW25113 with clsA, clsB and clsC genes deleted |
| genetic reagent (*E. coli*) | KEIO library | *Datsenko and Wanner, 2000* | | Nonessential genes disrupted in *E. coli* BW25113 |
| Recombinant DNA reagent | pKD4 | *Datsenko and Wanner, 2000* | Plasmid | Template for the amplification of a kanamycin resistance cassette flanked by FRT sites. |
| Recombinant DNA reagent | pKD46 | *Datsenko and Wanner, 2000* | Plasmid | Temperature sensitive, low copy number plasmid encoding the Lambda RED recombinase genes under the control of an arabinose inducible promoter |
| Recombinant DNA reagent | pCP20 | *Datsenko and Wanner, 2000* | Plasmid | Temperature sensitive plasmid encoding the FLP recombinase gene |

*Continued on next page*

*Continued*

| Reagent type (species) or resource | Designation | Source or reference | Identifiers | Additional information |
|---|---|---|---|---|
| Recombinant DNA reagent | pET17b | Novagen | Plasmid | T7 expression vector, AmpR |
| Recombinant DNA reagent | pET17b *dolP* | This paper | Plasmid | pET17b with *dolP* cloned between NdeI and EcoRI |
| Recombinant DNA reagent | pET17b *dolP* TM | This paper | Plasmid | As described above with the *dolP* gene randomly disrupted by Transposon mutations |
| Recombinant DNA reagent | pET17b *dolP* STm | This paper | Plasmid | pET17b with the *S. typhimurium dolP* gene cloned between NdeI and HindIII |
| Recombinant DNA reagent | pET17b *dolP* H.i | This paper | Plasmid | pET17b encoding a codon optimised *Haemophilus influenza dolP* homolog |
| Recombinant DNA reagent | pET17b *dolP* P.m | This paper | Plasmid | pET17b encoding a codon optimised *Pasteurella multocida dolP* homolog |
| Recombinant DNA reagent | pET17b *dolP* N.m | This paper | Plasmid | pET17b encoding a codon optimised *Neisseria meningitidis dolP* homolog |
| Recombinant DNA reagent | pET17b *dolP* V.c | This paper | Plasmid | pET17b encoding a codon optimised *Vibrio cholera dolP* homolog |
| Recombinant DNA reagent | pET17b *osmY* | This paper | Plasmid | pET17b encoding a codon optimised *E. coli* K12 *osmY* |
| Recombinant DNA reagent | p(OM)OsmY | This paper | Plasmid | pET17b encoding a codon optimised *E. coli* K12 *osmY* synthesised with the *dolP* signal sequence and acylation site in place of the *osmY* signal sequence |

*Continued on next page*

*Continued*

| Reagent type (species) or resource | Designation | Source or reference | Identifiers | Additional information |
|---|---|---|---|---|
| Recombinant DNA reagent | pET20b | Novagen | Plasmid | T7 expression vector, AmpR |
| Recombinant DNA reagent | pET20b *dolP* | This paper | Plasmid | pET20b with *dolP* cloned between NdeI and EcoRI |
| Recombinant DNA reagent | pET20b *dolP* PM | This paper | Plasmid | pET20b with *dolP* cloned between NdeI and EcoRI with site-directed point mutations at various sites |
| Recombinant DNA reagent | pET20b *wbbL* | This paper | Plasmid | pET20b with *wbbL* gene cloned between NdeI and HindIII |
| Recombinant DNA reagent | pET20b *dolP::mCherry* | This paper | Plasmid | pET20b encoding *dolP* fused to a codon optimised *mCherry* gene via a C-terminal 11-codon flexible linker (GGSSLVPSSDP) |
| Recombinant DNA reagent | pET26b *dolPpelB::mCherry* | This paper | Plasmid | pET26b *dolP::mCherry* with the *dolP* signal sequence replaced with that of *pelB* |
| Recombinant DNA reagent | pET20b *dolPIM::mCherry* | This paper | Plasmid | pET20b *dolP::mCherry* with codon 20 and 22 of *dolP* each mutated to aspartic acid |
| Recombinant DNA reagent | pET20b *dolPW127E::mCherry* | This paper | Plasmid | pET20b *dolP::mCherry* with codon 127 mutated to glutamic acid |

## Bioinformatic analyses

The BON-domain profile was obtained from Pfam http://pfam.sanger.ac.uk/ (*Punta et al., 2012*) and used as input for HMMER (hmmsearch version 3.1) (*Finn et al., 2011*) against the Uniprot database (http://www.uniprot.org, release 06032013) with an inclusion cutoff of E = 1 without heuristic filters. Sequence redundancy for clustering analysis was minimised using the UniRef100 resource of representative sequences; clustering was performed with the mclblastline program (*Enright et al., 2002*; *Hunter et al., 2012*) based on the e-value obtained by a BlastP run of all-against-all. Optimal settings for the mcl clustering were manually determined, clustering was performed at an e-value cutoff of 1E-2 and an inflation parameter of 1.2 using the scheme seven setting implemented in mcl. The resulting clusters were matched back to the proteins originally recovered by the HMMER search, and the number of proteins, as well as the number of matched organisms, are summarised for each phylum or subphylum in *Table 1*. UniProt accession numbers of all proteins according to their clusters are given in *Supplementary file 1*. The domain annotation was obtained from the InterPro database (*Hunter et al., 2012*). For cluster representation (*Figure 1*), the program CLANS (*Frickey and*

*Lupas, 2004*) was used under the default settings. Clusterings with CLANS was based on a subset of OsmY-, DolP- and Kbp-like proteins identified as described above; the respective accession numbers are given in *Table 4*. Pairwise alignment similarity values were analysed at the Protein Information Resource site (PIR; http://pir.georgetown.edu/).

## Plasmids, bacterial strains, and culture conditions

*Escherichia coli* BW25113 was the parental strain used for most investigations. *E. coli dolP::kan*, *osmY::kan* and *kbp::kan* mutants were obtained from the KEIO library (*Baba et al., 2006*) and the mutations transduced into a clean parental strain. *E. coli Δ dolP* was created by resolving the Kan$^R$ cassette, as previously described (*Datsenko and Wanner, 2000*). *E. coli* BW25113 *ΔpgsA* was constructed first by transfer of the *rcsF::aph* allele from the Keio library into *E. coli* BW25113 and removal of the *kan$^R$* cassette. The *lpp:aph* allele was then introduced into the ΔrcsF strain, and the cassette removed by the λ-Red recombination method of Datsenko and Wanner, due to the presence of Lpp being toxic in the absence of phosphatidylglycerol (*Datsenko and Wanner, 2000*; *Kikuchi et al., 2000*; *Suzuki et al., 2002*). Finally, the same method was utilised to create the *ΔpgsA* strain (*ΔrcsF,Δlpp,ΔpgsA*) The genes encoding DolP and OsmY were amplified from *E. coli* BW25113 and cloned into pET17b to create pDolP and pOsmY. Orthologous sequences from *S. enterica, V. cholera, N. meningitidis, H. influenza* and *P. multocida* were synthesised and cloned into pET17b to create the plasmids pSe, pVc, pNm, pHi, and pPm, respectively. To create pDolP$^{pelB}$, the gene encoding DolP was synthesised but with nucleotides encoding the PelB signal sequence in place of the native signal sequence and without Cys19 to relieve the possibility of acylation; this plasmid was constructed in pET26b+ such that the protein had a C-terminal His-tag. In addition, to create p(OM) OsmY the gene encoding OsmY was synthesised but with nucleotides encoding the native DolP signal sequence and Cys19 N-terminal acylation site in place of the native OsmY signal sequence. The latter plasmid was constructed in pET17b. The pET17b-*dolP::mCherry* plasmid was constructed to contain an 11 amino acid flexible linker and a codon optimised mCherry gene at the 3' end of the *dolP* gene. Gene synthesis was performed by Genscript. The pet20b+-*wbbL* plasmid for restoring O-antigen synthesis in *E. coli* K-12 was previously described (*Browning et al., 2013a*). Single point mutations were generated by using Quickchange II according to manufacturer's instructions. All constructs were confirmed by DNA sequencing. Strains were routinely cultured on LB agar and LB broth. Linker scanning mutagenesis was performed with an Ez-Tn5 kit (Epicentre) as previously described (*Browning et al., 2013b*).

**Table 4.** Accession numbers for the sequences used for CLANS clustering shown in *Figure 1*.

| Organism | OsmY | DolP | Kbp |
|---|---|---|---|
| *Escherichia coli* K12 | P0AFH8 | P64596 | P0ADE6 |
| *Klebsiella pneumoniae* MGH 78578 | A6THZ1 | A6TEG9 | A6T985 |
| *Enterobacter cloacae* ENHKU01 | J7G7C8 | J7GHD1 | J7GFT3 |
| *Salmonella enterica* Typhimurium | Q7CP68 | Q7CPQ6 | Q8ZML9 |
| *Erwinia billingiae* Eb661 | D8MMS8 | D8MME2 | D8MNV6 |
| *Serratia proteamaculans* 568 | A8G9G9 | A8GJZ3 | A8GFP7 |
| *Cronobacter sakazakii* ATCC BAA-894 | A7MGB6 | A7MIQ1 | A7MEA9 |
| *Pantoea* sp. Sc1 | H8DPK0 | H8DQ90 | H8DIH9 |
| *Hafnia alvei* ATCC 51873 | G9Y3J7 | G9Y4J4 | G9YAM4 |
| *Citrobacter rodentium* ICC168 | D2TRY4 | D2TQ24 | D2TM58 |
| *Shigella flexneri* 1235–66 | I6F1Q5 | I6GLP1 | I6HD15 |
| *Yersinia enterocolitica* 8081 | A1JJ93 | A1JR75 | |
| *Yersinia pestis* KIM10+ | Q7CG58 | Q8D1R6 | |
| *Dickeya dadantii* 3937 | E0SJX0 | E0SHF6 | |

## Analysis of membrane lipid content

Cell envelopes of *E. coli* were separated by defined sucrose density gradient separation, precisely as described previously following cell disruption by 3 passes of the C3 emulsiflex (Avestin) (*Isom et al., 2017*; *Dalebroux et al., 2015*). Samples were generated in biological triplicate from three separate 2 L batches of cells grown to an $OD_{600}$ 0.6–0.8, with the final volumes for washed membranes being 1 ml, which were stored at −80 ℃ until analysis. Lipids were extracted by the Bligh-Dyer method (*Bligh and Dyer, 1959*) from purified membranes as described previously (*Isom et al., 2017*). Methanol and chloroform were added to the samples to extract the metabolites using a modified Bligh-Dyer procedure (*Wu et al., 2008*) with a final methanol/chloroform/water ratio of 2:2:1.8. The non-polar layer was extracted and dried under nitrogen before being stored at −80 ℃ until analysis. Samples were re-dissolved in 200 µl chloroform before being separated by thin layer chromatography on silica gel 60 plates with the mobile phase as chloroform:methanol:water at the following ratio: 65:25:10. Lipids were visualised by staining with phosphomolybdic acid. Analysis of lipid samples by mass spectrometry was completed as described previously (*Teo et al., 2019*). The differences were as follows: lipid extracts were diluted 10x or 20x into starting LC solvent the LC-MS/MS run directly. Normalisation was completed by taking the ion intensity of each phospholipid relative to the total ion count.

## Biochemical analyses

Cellular fractions were prepared as described previously (*Parham et al., 2004*). Cellular fractions and purified proteins were electrophoresed on 12 or 15% SDS-PAGE gels and stained with Coomassie blue or transferred to a polyvinylidene difluoride (PVDF) membrane for Western immunoblotting as previously described (*Leyton et al., 2011*). Loading consistency was confirmed by immuno-blotting with anti-BamB or anti-PqiB antiserum where possible. Protease shaving assays were described previously (*Selkrig et al., 2012*). Proteins were localised by immunofluorescence as described previously (*Leyton et al., 2011*). Analytical ultracentrifugation was performed as described previously (*Knowles et al., 2011*). For proteomic analysis of OM protein content, OM fractions purified by defined sucrose gradient centrifugation in biological triplicate and were digested with trypsin using the FASP method (*Wiśniewski et al., 2009*). Primary amines in the peptides were then dimethylated using hydrogenated or deuterated formaldehyde and sodium cyanoborohydride. Labelled peptides were mixed, separated into 15 fractions by mixed-mode reverse-phase/anion exchange chromatography, the fractions lyophilised and each analysed with a 90 min LC-MS/MS run using a Bruker Impact Q-TOF mass spectrometer. Data was searched against forward and randomised *E. coli* sequence databases using MASCOT and filtered at 1% FDR. Quantitation was based on the extracted ion chromatograms of light/heavy peptide pairs. DolP was investigated for binding partners using immunoprecipitation assays as described previously. Briefly, *E. coli* Δ*dolP*, and isogenic strains containing pDolP[pelB] or plasmid containing a His-Tagged version of BamA were grown in LB media to an $OD_{600}$ of ~0.6 and harvested by centrifugation. Cells were resuspended in PBS with Triton X-100 supplemented with lysozyme and Benzonase nuclease. Cells were lysed and clarified by centrifugation. The lysate was incubated with Ni-NTA agarose (Qiagen) or appropriate antibodies. Precipitated proteins were analysed by Western immunoblotting.

## NMR spectroscopy

Experiments were carried out at 298 K on a Varian Inova 800 MHz spectrometer equipped with a triple-resonance cryogenic probe and *z*-axis pulse-field gradients. Isotope labelled DolP ($^{15}$N $^{13}$C) with its N-terminal cysteine replaced was used at a concentration of 1.5 mM in 50 mM sodium phosphate (pH 6), 50 mM NaCl and 0.02% $NaN_3$ in 90% $H_2O$/10% $D_2O$. Spin system and sequential assignments were made from CBCA(CO)NH, HNCACB, HNCA, HN(CO)CA, HNCO, HN(CA)CO, H(C)CH TOCSY and (H)CCH TOCSY experiments (*Muhandiram and Kay, 1994*). Spectra were processed with NMRPipe (*Delaglio et al., 1995*) and analysed with SPARKY (*Goddard and Kneller, 2008*).

## Structure calculations

Interproton distance restraints were obtained from $^{15}$N- and $^{13}$C-edited NOESY-HSQC spectra ($\tau_{mix}$=100 ms). PRE restraints were obtained by adding 10 mM DPC/3.33 mM CHAPS micelles spiked with 1 mM DMPG and 0.185 mM 5-doxyl 1-palmitoyl-2-steroyl-sn-glycero-phosphocholine (Avanti,

Polar Lipids, Alabaster, AL, USA) to $^{15}$N-labelled DolP (300 μM) and by standardising amide resonance intensities to those induced by spiking instead with unlabelled dipalmitoyl phosphocholine (Avanti Polar Lipids). Backbone dihedral angle restraints (φ and ψ) were obtained using TALOS from the backbone chemical shifts (*Cornilescu et al., 1999*). Slowly exchanging amides were deduced from the $^1$H $^{15}$N SOFAST-HSQC (*Schanda et al., 2005*) spectra of protein dissolved in 99.96% D$_2$O. The structure was calculated iteratively using CANDID/CYANA, with automated NOE cross-peak assignment and torsion angle dynamics implemented (*Güntert, 2004*). A total of 20 conformers with the lowest CYANA target function were produced that satisfied all measured restraints. Aria1.2 was used to perform the final water minimisation (*Linge et al., 2001*). Structures were analysed using PROCHECK-NMR (*Laskowski et al., 1993*) and MOLMOL (*Koradi et al., 1996*). Structural statistics are summarised in *Table 2*.

### Lipid interactions

Ligand binding to 300 μM $^{15}$N- DolP in 50 mM sodium phosphate (pH 6), 50 mM NaCl and 0.02% NaN$_3$ in 90% H$_2$O/10% D$_2$O was monitored by $^1$H$^{15}$N-HSQCs at concentrations of 0–40 mM of either DHPG or DHPE (c.m.c.,~7 mM). The DPC-DMPG: DolP complex was calculated by HADDOCK (*Dominguez et al., 2003*; *Dancea et al., 2008*). A total of 18 paramagnetic relaxation enhancements restrained the distances between the micelle centre and the respective NH groups to 0–20 Å, with CSPs defining the flexible zone. The top 200 models were ranked according to their experimental energies and statistics derived from the 20 lowest-energy conformers were reported (*Table 5*).

### Small-angle X-ray scattering

Synchrotron SAXS data of DolP were collected at the EMBL X33 beamline (DESY, Hamburg) using a robotic sample changer. DolP concentrations between 1 and 10 mg/ml were run in 50 mM sodium phosphate (pH 6), 50 mM NaCl and 0.02% NaN$_3$. Data were recorded on a PILATUS 1M pixel detector (DECTRIS, Baden, Switzerland) at a sample-detector distance of 2.7 m and a wavelength of 1.5 Å, covering a range of momentum transfer of $0.012 < s < 0.6$ Å$^{-1}$ ($s = 4\pi\sin(\theta)/\gamma$, where 2θ is the scattering angle) and processed by PRIMUS (*Konarev et al., 2003*). The forward scattering I(0) and the radius of gyration (R$_g$) were calculated using the Guinier approximation (*Guinier, 1939*; *Figure 2—figure supplement 6*). The pair-distance distribution function pR, from which the maximum particle dimension (D$_{max}$) is estimated, was computed using GNOM (*Svergun, 1992*; *Figure 2—figure supplement 6*). Low resolution shape analysis of the solute was performed using DAMMIF (*Franke and Svergun, 2009*). Several independent simulated annealing runs were performed and the results were analysed using DAMAVER (*Volkov and Svergun, 2003*). Back comparison of the DolP solution structure with the SAXS data was performed using the ensemble optimisation method

**Table 5.** HADDOCK docking statistics for ensemble 20 lowest-energy DolP-DPC micelle solution structures calculated.

| Experimental parameters[*] | |
| --- | --- |
| Ambiguous distance restraints | 19 including NH of I20, G120-T130, V132-Q135, T138, S139, and NHε of W127 |
| Number of flexible residues[†] | 50 (I20-V45 (flexible linker as ascertained by NMR), A74, G120-I128, K131-R133, Q135-L137, V142-S145, I173,S178-V180) |
| Atomic pairwise RMSD (Å) | |
| All backbone | |
| Flexible interface backbone | |
| Intermolecular energies (kcal. mol$^{-1}$) | |
| E$_{vdw}$ | −100.81 ± 7.74 |
| E$_{elec}$ | −231.67 ± 64.14 |
| E$_{restraints}$ | 22.30 ± 4.29 |
| Buried surface area (Å$^2$) | 2186.78 ± 133.277 |

* deduced from intensity reductions observed in presence of 5-doxl derivative.

† according to their surface accessibility and the chemical shift perturbation in presence of DPC/CHAPS.

(*Bernadó et al., 2008*) accounting for flexibility between residues 20–46, 112–118 and 189–195. All programs used for analysis of the SAXS data belong to the ATSAS package (*Petoukhov and Svergun, 2005*).

## Accession codes

Coordinates and NMR assignments have been deposited with accession codes 7A2D (PDB) and 19760 (BMRB), respectively.

## Cell imaging

Cultures were grown at 37°C to $OD_{600}$0.4–0.5. Cells were harvested by centrifugation at 7000 x g for 1 min before being applied to agarose pads, which were prepared with 1.5% agarose in PBS and set in Gene Frames (Thermo Scientific). Cells were immediately imaged using a Zeiss AxioObserver equipped with a Plan-Apochromat 100x/Oil Ph3 objective and illumination from HXP 120V for phase contrast images. Fluorescence images were captured using the Zeiss filter set 45, with excitation at 560/40 nm and emission recorded with a bandpass filter at 630/75 nm. For localisation analysis and generation of demographs, the MicrobeJ plugin for Fiji was used and >500 cells were used as input for analysis (*Ducret et al., 2016*).

## Membrane fluidity assay

Membrane fluidity was measured by use of the membrane fluidity assay kit (Abcam: ab189819) as was described previously except with minor modifications (*Storek et al., 2019*). Specific bacterial strains were grown to stationary phase overnight (~16 hr) after which cells were harvested by centrifugation, washed with PBS three times and finally labelled with labelling mix (10 µM pyrenedecanoic acid and 0.08% pluronic F-127 in PBS) for 20 min in the dark at 25°C with shaking. Cells were washed twice with PBS before fluorescence was recorded with excitation at 350 nm and emission at either 400 nm or 470 nm to detect emission of the monomer or excimer respectively. Unlabelled cells were used as a control to confirm labelling and the *E. coli* BW25113 Δ*waaD* strain was used as a positive control for increased membrane fluidity. Following subtraction of fluorescence from the blanks, averages from triplicate experiments were used to calculate the ratio of excimer to monomer fluorescence. These ratios were then expressed as relative to the parent *E. coli* BW25113 strain.

## Genetic interaction analysis

Genetic interaction assay was performed as described in *Banzhaf et al., 2020*. For each probed strain, a single source plate was generated and transferred to the genetic interaction plate using a pinning robot (Biomatrix 6). On each genetic interaction assay plate, the parental strain, the single deletion A, the single deletion B and the double deletion AB were arrayed, each in 96 copies per plate. Genetic interaction plates were incubated at 37°C for 12 hr and imaged under controlled lighting conditions (spImager S and P Robotics) using an 18-megapixel Canon Rebel T3i (Canon). Colony integral opacity as fitness readout was quantified using the image analysis software Iris (*Kritikos et al., 2017*). Fitness ratios were calculated for all mutants by dividing their fitness values by the respective WT fitness value. The product of single mutant fitness ratios (expected) was compared to the double mutant fitness ratio (observed) across replicates. The probability that the two means (expected and observed) are equal across replicates is obtained by a Student's two-sample *t*-test.

## Lipid A palmitoylation assay

Labelling of LPS, Lipid A purification, TLC analysis, and quantification were done exactly as described previously (*Chong et al., 2015*). The positive control was exposed to 25 mM EDTA for 10 min prior to harvest of cells by centrifugation in order to induce PagP mediated palmitoylation of Lipid A (*Chong et al., 2015*). Experiments were completed in triplicate and the data generated was analysed as described previously.

## Acknowledgements

This work was supported by the BBSRC (IRH and MO: BB/M00810X/1 and BB/L00335X/1; TJK BB/P009840/1), NSERC RGPIN-2018–04994 and Campus Alberta Innovation Program (RCP-12–002C) (MO). We would like to thank Georgia L Isom and Catherine A Wardius for technical assistance in the laboratory. We thank Professor Corinne Spickett for use of mass spectrometry facilities for phospholipid analyses. We also thank Professor Jeff Cole for critical advice in development of the project.

## Additional information

### Funding

| Funder | Grant reference number | Author |
|---|---|---|
| Biotechnology and Biological Sciences Research Council | BB/M00810X/1 | Michael Overduin Ian R Henderson |
| Biotechnology and Biological Sciences Research Council | BB/L00335X/1 | Michael Overduin Ian R Henderson |
| Biotechnology and Biological Sciences Research Council | BB/P009840/1 | Timothy J Knowles |
| Campus Alberta Neuroscience | RCP-12-002C | Michael Overduin |
| Natural Sciences and Engineering Research Council of Canada | RGPIN-2018–04994 | Michael Overduin |

The funders had no role in study design, data collection and interpretation, or the decision to submit the work for publication.

### Author contributions

Jack Alfred Bryant, Conceptualization, Data curation, Formal analysis, Supervision, Validation, Investigation, Visualization, Methodology, Writing - original draft, Project administration, Writing - review and editing; Faye C Morris, Conceptualization, Data curation, Formal analysis, Investigation, Methodology, Writing - review and editing; Timothy J Knowles, Conceptualization, Resources, Data curation, Formal analysis, Supervision, Investigation, Methodology, Writing - review and editing; Riyaz Maderbocus, Data curation, Formal analysis, Investigation; Eva Heinz, Software, Formal analysis, Investigation, Methodology; Gabriela Boelter, Dema Alodaini, Adam Colyer, Peter J Wotherspoon, Kara A Staunton, Mark Jeeves, Douglas F Browning, Yanina R Sevastsyanovich, Timothy J Wells, Amanda E Rossiter, Vassiliy N Bavro, Pooja Sridhar, Douglas G Ward, Zhi-Soon Chong, Emily CA Goodall, Christopher Icke, Alvin CK Teo, Investigation; Shu-Sin Chng, Supervision, Validation; David I Roper, Trevor Lithgow, Adam F Cunningham, Supervision, Validation, Writing - review and editing; Manuel Banzhaf, Software, Supervision, Validation, Investigation, Writing - review and editing; Michael Overduin, Conceptualization, Resources, Supervision, Funding acquisition, Validation, Project administration, Writing - review and editing; Ian R Henderson, Conceptualization, Resources, Supervision, Funding acquisition, Validation, Writing - original draft, Project administration, Writing - review and editing

### Author ORCIDs

Jack Alfred Bryant ⓘ https://orcid.org/0000-0002-7912-2144
Faye C Morris ⓘ https://orcid.org/0000-0002-9021-0452
Eva Heinz ⓘ https://orcid.org/0000-0003-4413-3756
Emily CA Goodall ⓘ https://orcid.org/0000-0003-4846-6566
Christopher Icke ⓘ https://orcid.org/0000-0002-7815-8591
Shu-Sin Chng ⓘ https://orcid.org/0000-0001-5466-7183
Ian R Henderson ⓘ https://orcid.org/0000-0002-9954-4977

Decision letter and Author response
Decision letter https://doi.org/10.7554/eLife.62614.sa1
Author response https://doi.org/10.7554/eLife.62614.sa2

# Additional files

## Supplementary files

• Supplementary file 1. UniProt accession numbers for the proteins in the respective clusters as shown in *Table 1*.

• Supplementary file 2. Mass spectrometry of outer membrane fractions to assess presence of protein.

• Transparent reporting form

## Data availability

Structural data have been deposited in PDB under the accession code 7A2D and the BMRB 19760. All data generated or analysed during this study are included in the manuscript and supporting files. We have supplied original images in the source data as appropriate. We have also supplied input data for Figure 1B in Supplementary file 1 and raw data for mass spectrometry results in Supplementary file 2.

The following datasets were generated:

| Author(s) | Year | Dataset title | Dataset URL | Database and Identifier |
|---|---|---|---|---|
| Bryant JA, Morris FC, Knowles TJ, Maderbocus R, Heinz E, Boelter G, Alodaini D, Colyer A, Wotherspoon PJ, Staunton KA, Jeeves M, Browning DF, Sevastsyanovich YR, Wells TJ, Rossiter AE, Bavro VN, Sridhar P, Ward DG, Chong ZS, Goodall ECA, Icke C, Teo ACK, Chng SS, Roper DI, Lithgow T, Cunningham AF, Banzhaf M, Overduin M, Henderson IR | 2020 | Structure of dual-BONdomain protein DolP identifies phospholipid binding as a new mechanism for protein localization | https://www.rcsb.org/structure/7A2D | RCSB Protein Data Bank, 7A2D |
| Bryant JA, Morris FC, Knowles TJ, Maderbocus R, Heinz E, Boelter G, Alodaini D, Colyer A, Wotherspoon PJ, Staunton KA, Jeeves M, Browning DF, Sevastsyanovich YR, Wells TJ, Rossiter AE, Bavro VN, Sridhar P, Ward DG, Chong ZS, Goodall ECA, Icke C, Teo ACK, Chng SS, Roper DI, Lithgow T, Cunningham AF, | 2020 | Structure-function analyses of dual-BON domain protein DolP identifies phospholipid binding as a new mechanism for protein localisation to the cell division site | http://www.bmrb.wisc.edu/data_library/summary/index.php?bmrbId=19760 | Biological Magnetic Resonance Data Bank, bmrbId=19760 |

Banzhaf M,
Overduin M,
Henderson IR

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
