## [Decision Letter]

**Acceptance summary:**

These studies investigate the structure, interactions, and localization of DolP, a BON domain-containing protein that localizes to the outer membrane. It is shown that DolP binds anionic phospholipids in a novel manner which then guides the protein to the cell division site. These studies are an important contribution to our understanding of outer membrane biogenesis and a large step forward in understanding Bon domain containing proteins. Furthermore, DolP might be exploited as a potential target for therapy against Gram-negative organisms.

**Decision letter after peer review:**

Thank you for submitting your work entitled "Structure of dual-BON domain protein DolP identifies phospholipid binding as a new mechanism for protein localization" for consideration by *eLife*. Your article has been reviewed by three peer reviewers, one of whom is a member of our Board of Reviewing Editors, and the evaluation has been overseen by Olga Boudker as the Senior Editor.

The reviewers are very positive about your manuscript, and have only minor suggestions which you might address before acceptance:

Reviewer #1:

The authors study the function of DolP, a dual-BON domain protein that is required for outer membrane integrity. Previous protein:protein interaction data suggested an interaction between DolP and components of the BAM complex in the outer membrane. But even though negative genetic interactions were observed between *dolP* and *bamB* and *bamE* no significant interactions could be detected between through immunoprecipitations. Therefore the authors set-out to perform a very extensive study of DolP, where they combine genetics, cell biology, structural biology and molecular biology to determine the structure of DolP and to reveal a novel mechanism by which DolP binds to anionic phospholipids via its BON2 domain which then guides the protein to the cell division site. The study is very clearly described and reveals many aspects of DolP. Although the authors are unable to find a direct mechanism through which DolP maintains OM integrity, this manuscript provides a significant scientific progress. I have no further major comments.

At some positions in the manuscript the authors could add some sentences to aid a reader coming from a different field. For example the authors could shortly explain the function of polymxin B in Figure 1—figure supplement 1 or describe the method to detect negative genetic interactions in some more detail.

Reviewer #2:

This work by Bryant et al. uses biophysical and cell biology methods to investigate the structure, interactions, and localization of DolP, an outer membrane (OM) protein in gram-negative bacteria that plays a role in determining the integrity of the OM. Overall the manuscript is well written and the results important. Indeed, although the exact mechanisms by which DolP maintains integrity of the OM were not determined, the atomic resolution structure of DolP and of its complex with a phospholipid micelle reported here provide the basis for antimicrobial design and deserve publication in *eLife*. However, some minor issues related to the characterization of the structure and phospholipid binding properties of DolP need to be addressed before publication.

1) It is important to specify how many interdomain NOEs were used in the refinement.

2) The authors mention that the two domains are hold in position via interdomain contacts mediated by Y75 and V82 in BON1 and T150, G160, L161 and T188 in BON2. Are these contacts evident from the NMR spectra? It would be appropriate to show NOEs among these residues in a figure.

3) When displaying the SAXS data in Figure 2, it is important to show the residual plot and to zoom in the low-q region (q < 0.2 A^-1). This is the region of the SAXS curve that is more sensitive to domain/domain orientation.

4) In Figure 3, it would be important to show the CSP versus phospholipid concentration plot that was used to fit a Kd ~ 80 uM. Why CSPs were measured at 40 mM DHPG if the Kd is 80 uM? Please resolve the ambiguity between surfactant and micelle concentration in the text.

Reviewer #3:

The manuscript of Bryant et al., describes the role and function of the formerly designated YraP protein, renaming it DolP, in *Escherichia coli*. This protein has been known to affect several cellular processes, but until now, there has been no unequivocal function assigned to it. In addition, the authors have solved the NMR structure of the protein (the first for a BON domain containing protein), which adds to a greater understanding of the functionality of DolP and importantly adds to our understanding of maintenance of outer membrane integrity in Gram-negative organisms. Moreover, the authors point out the potential for DolP to be exploited as a potential target for therapy against Gram-negative organisms.

DolP is a BON domain containing protein, which are highly conserved in Gram-negative bacteria, and appear to bind anionic phospholipids in a novel manner and where by lipid binding is essential for DolP function. The complementation experiments with the PelB signal sequence are elegant and demonstrate that the covalent linkage via C19 is required for functionality.

Overall the paper is well written and is an important contribution to our understanding of *E. coli* OM biogenesis and a large step forward in BON protein understanding. In principle I would support publication of this in *eLife* given the importance to the field. The conclusions are supported by the data in the manuscript. I do not believe that further work is required, although some clarification of a few points is required by the authors.

Define BON as this does not appear to be defined – 'bacterial OsmY and nodulation'.

The DolP and BamB and BamE interaction work needs more explanation, especially the link to membrane fluidity…maybe I missed something…but seems to be a jump of logic I don't follow- is there interaction with other members of Bam?

In the Discussion, it would be nice if the authors could add some commentary on how the DolP system may function in concert with systems that are involved in lipid transport and maintenance of the OM such as the Mla system.

Figure 1—figure supplement 4B – Growth Curve should be a semi-log plot.

---

## [Author Response]

Reviewer #1:The authors study the function of DolP, a dual-BON domain protein that is required for outer membrane integrity. Previous protein:protein interaction data suggested an interaction between DolP and components of the BAM complex in the outer membrane. But even though negative genetic interactions were observed between dolP and bamB and bamE no significant interactions could be detected between through immunoprecipitations. Therefore the authors set-out to perform a very extensive study of DolP, where they combine genetics, cell biology, structural biology and molecular biology to determine the structure of DolP and to reveal a novel mechanism by which DolP binds to anionic phospholipids via its BON2 domain which then guides the protein to the cell division site. The study is very clearly described and reveals many aspects of DolP. Although the authors are unable to find a direct mechanism through which DolP maintains OM integrity, this manuscript provides a significant scientific progress. I have no further major comments.At some positions in the manuscript the authors could add some sentences to aid a reader coming from a different field. For example the authors could shortly explain the function of polymxin B in Figure 1—figure supplement 1 or describe the method to detect negative genetic interactions in some more detail.

We added the following text to the figure legend for Figure 1—figure supplement 1:

“polymyxin B, which permeabilizes the OM, allowing the protease access to the periplasm.”

We added the following text to the figure legend for Figure 3—figure supplement 1:

“Fitness was measured by quantifying colony size and integral opacity, which represents colony density, using the image analysis software Iris (Kritikos et al., 2017).”

Reviewer #2:This work by Bryant et al. uses biophysical and cell biology methods to investigate the structure, interactions, and localization of DolP, an outer membrane (OM) protein in gram-negative bacteria that plays a role in determining the integrity of the OM. Overall the manuscript is well written and the results important. Indeed, although the exact mechanisms by which DolP maintains integrity of the OM were not determined, the atomic resolution structure of DolP and of its complex with a phospholipid micelle reported here provide the basis for antimicrobial design and deserve publication in eLife. However, some minor issues related to the characterization of the structure and phospholipid binding properties of DolP need to be addressed before publication.1) It is important to specify how many interdomain NOEs were used in the refinement.

We have added this information in the revised and updated statistics Table 2, subsection “The structure of DolP reveals a dual-BON domain lipoprotein”, Figure 2D legend, Figure 2—figure supplement 5 and Table 3.

2) The authors mention that the two domains are hold in position via interdomain contacts mediated by Y75 and V82 in BON1 and T150, G160, L161 and T188 in BON2. Are these contacts evident from the NMR spectra? It would be appropriate to show NOEs among these residues in a figure.

We have moved the structural representation from Figure 2D to Figure 2—figure supplement 5 to show more clearly all NOEs between interdomain contact residues and have listed these NOEs in Table 3.

3) When displaying the SAXS data in Figure 2, it is important to show the residual plot and to zoom in the low-q region (q < 0.2 A^-1). This is the region of the SAXS curve that is more sensitive to domain/domain orientation.

We have included Figure 2—figure supplement 4 to show the zoom of the low q region and the residuals plots as requested.

4) In Figure 3, it would be important to show the CSP versus phospholipid concentration plot that was used to fit a Kd ~ 80 uM. Why CSPs were measured at 40 mM DHPG if the Kd is 80 uM? Please resolve the ambiguity between surfactant and micelle concentration in the text.

We thank the reviewer for pointing out this error and have repeated the analysis and included the relevant plots in Figure 3—figure supplement 4. We have resolved the ambiguity in the subsection “DolP binds specifically to anionic phospholipids via BON2”.

Reviewer #3:The manuscript of Bryant et al., describes the role and function of the formerly designated YraP protein, renaming it DolP, in *Escherichia coli*. This protein has been known to affect several cellular processes, but until now, there has been no unequivocal function assigned to it. In addition, the authors have solved the NMR structure of the protein (the first for a BON domain containing protein), which adds to a greater understanding of the functionality of DolP and importantly adds to our understanding of maintenance of outer membrane integrity in Gram-negative organisms. Moreover, the authors point out the potential for DolP to be exploited as a potential target for therapy against Gram-negative organisms.DolP is a BON domain containing protein, which are highly conserved in Gram-negative bacteria, and appear to bind anionic phospholipids in a novel manner and where by lipid binding is essential for DolP function. The complementation experiments with the PelB signal sequence are elegant and demonstrate that the covalent linkage via C19 is required for functionality.Overall the paper is well written and is an important contribution to our understanding of *E. coli* OM biogenesis and a large step forward in BON protein understanding. In principle I would support publication of this in eLife given the importance to the field. The conclusions are supported by the data in the manuscript. I do not believe that further work is required, although some clarification of a few points is required by the authors.Define BON as this does not appear to be defined – 'bacterial OsmY and nodulation'.

Now defined in the Introduction at first usage in the main text.

The DolP and BamB and BamE interaction work needs more explanation, especially the link to membrane fluidity…maybe I missed something…but seems to be a jump of logic I don't follow- is there interaction with other members of Bam?

In the Discussion and in the Results we highlight that previous protein:protein interaction studies highlighted DolP as a potential near neighbour of the Bam complex components BamD and BamE.

We added the following to explain the reasoning for the genetic interaction analysis:

“As the loss of multiple genes encoding different components of a single pathway can have additive phenotypes, such as decreased fitness, we investigated strains with dual mutations in *dolP* and genes coding the non-essential BAM complex components *bamB* or *bamE*”

We also added the following to explain the genetic link between *dolP* and *bamB/ba*m*E*:

“Considering that *bamB* mutants are sensitive to increased membrane fluidity (Storek et al., 2019), these data suggest that the genetic interaction between *dolP* and *bamE* or *bamB*, observed here, is likely facilitated indirectly through changes to membrane fluidity on the loss of DolP”

In the Discussion, it would be nice if the authors could add some commentary on how the DolP system may function in concert with systems that are involved in lipid transport and maintenance of the OM such as the Mla system.

We have added the following line to the Discussion:

“No differences in LPS content or OM asymmetry were observed in a *dolP* mutant suggesting DolP does not influence the OM phospholipid recycling Mla pathway or LPS biogenesis”

Figure 1—figure supplement 4B – Growth Curve should be a semi-log plot

This has now been amended.